DOI: 10.1038/s41467-017-01829-1 | OPEN

# The IKK/NF-κB signaling pathway requires Morgana to drive breast cancer metastasis

Federica Fusella[1], Laura Seclì[1], Elena Busso[1], Anna Krepelova[2,3], Enrico Moiso [1], Stefania Rocca[1], Laura Conti[1], Laura Annaratone [4], Cristina Rubinetto[1], Maurizia Mello-Grand[5], Vijay Singh [5,6], Giovanna Chiorino[5], Lorenzo Silengo[1], Fiorella Altruda[1], Emilia Turco[1], Alessandro Morotti [7], Salvatore Oliviero[2,3], Isabella Castellano[4], Federica Cavallo [1], Paolo Provero [1,8], Guido Tarone[1] & Mara Brancaccio[1]

NF-κB is a transcription factor involved in the regulation of multiple physiological and pathological cellular processes, including inflammation, cell survival, proliferation, and cancer cell metastasis. NF-κB is frequently hyperactivated in several cancers, including triple-negative breast cancer. Here we show that NF-κB activation in breast cancer cells depends on the presence of the CHORDC1 gene product Morgana, a previously unknown component of the IKK complex and essential for IκBα substrate recognition. Morgana silencing blocks metastasis formation in breast cancer mouse models and this phenotype is reverted by IκBα downregulation. High Morgana expression levels in cancer cells decrease recruitment of natural killer cells in the first phases of tumor growth and induce the expression of cytokines able to attract neutrophils in the primary tumor, as well as in the pre-metastatic lungs, fueling cancer metastasis. In accordance, high Morgana levels positively correlate with NF-κB target gene expression and poor prognosis in human patients.

[1] Department of Molecular Biotechnology and Health Sciences, University of Torino, Torino 10126, Italy. [2] Department of Life Sciences and Systems Biology, University of Torino, Via Accademia Albertina 13, Torino 10123, Italy. [3] Human Genetics Foundation (HuGeF), Via Nizza 52, Torino 10126, Italy. [4] Department of Medical Sciences, University of Torino, Torino 10126, Italy. [5] Cancer Genomics Laboratory, Fondazione Edo ed Elvo Tempia, Biella 13900, Italy. [6] Centre for Biological Sciences, Central University of Bihar, Patna Campus, Bihar 800014, India. [7] Department of Clinical and Biological Sciences, University of Torino, Orbassano 10043, Italy. [8] Center for Translational Genomics and Bioinformatics, San Raffaele Scientific Institute, Milan 20132, Italy. Federica Fusella and Laura Seclì contributed equally to this work. Correspondence and requests for materials should be addressed to F.F. (email: federica.fusella@unito.it) or to M.B. (email: mara.brancaccio@unito.it)

**B**reast carcinoma is a common malignancy in woman. Despite the recent advance in diagnostic methods, the mortality for this cancer remains high. Patient death results entirely from metastasis formation in distant organs. Definitively, our possibility to control the pathology depends on our ability to block cancer metastasis by understanding the subtended molecular mechanisms. This task is made difficult by the complexity of the metastatic process. Cancer metastasis is a multistage event, beginning with local invasion, vessel intravasation, survival in suspension, extravasation in foreign organs and re-entering in the cell cycle[1]. It is conceivable that tuning of hundred of genes is necessary to achieve the task.

The family of NF-κB transcription factors consists of five members, RelA/p65, c-Rel, RelB, NF-κB1 (p50), and NF-κB2 (p52), forming homo- and heterodimers. In absence of specific stimuli, these dimers are bound to IκB (inhibitors of NF-κB) and are kept transcriptionally inactive. The NF-κB signaling pathway responds to different stimuli, from cytokine and growth factor signaling to the recognition of pathogen products or DNA damages and oncogenic stress. The activation of the NF-κB "canonical pathway" induces the formation of the IKK (IκB kinase) protein complex containing the IKKα and IKKβ kinases and the regulatory subunit IKKγ/NEMO. The activated IKK complex phosphorylates IκBα inducing its detachment from NF-κB and its degradation. As a result, the NF-κB dimers can enter the nucleus and regulate the transcription of their target genes[2–4]. NF-κB is involved in activating immune and inflammatory responses but also in regulating adhesion, angiogenesis, autophagy, energy metabolism, senescence, and inducing cell proliferation and survival[5–7]. It is therefore not surprising that NF-κB has been involved in cancer onset and progression both in experimental models and in human patients[8–11]. Aberrant activation of NF-κB is frequently found in triple-negative breast cancer (TNBC)[12, 13]. This cancer subtype accounts for 15–20% of all breast cancers and it is characterized by high rate of recurrence and poor prognosis. TNBC does not express estrogen receptors, progesterone receptors and lacks HER-2 overexpression. As a consequence, women with TNBC do not benefit from endocrine therapy or treatment with monoclonal antibodies and tyrosine kinase inhibitors against HER-2. For these reasons, identifying innovative treatments for TNBC is an urgent need.

We recently identified Morgana/chp-1, coded by the CHORDC1 gene, as a protein overexpressed in 36% of TNBC[14]. Morgana is a ubiquitously expressed protein[15, 16] with chaperone activity[17, 18], essential for mouse and Drosophila development. Morgana binds to Hsp90 chaperone protein[17–22], behaving as a co-chaperone[17] and interacts with and inhibits Rho kinases I and II[14, 20, 23]. Morgana displays oncogenic features, conferring resistance to apoptosis when overexpressed[24]. Indeed, high Morgana expression levels by excessively inhibiting ROCK I activity, destabilize PTEN, hence triggering the PI3K/AKT survival pathway[14]. Given that apoptosis resistance is an important feature of metastatic cells and that Morgana overexpression correlates with lymph node positivity in breast cancer patients[14], we decided to investigate the role of Morgana in breast cancer metastasis. We show that Morgana is an essential component of the IKK complex, required for TNBC cell invasion in vitro and in vivo, independently of ROCK activity. Morgana overexpression potently sustains the NF-κB signaling pathway, leading to pro-metastatic gene expression and cancer cell invasion. Experimental and in silico gene expression analyzes on breast cancer patients confirm that Morgana correlates with NF-κB target gene expression and with poor survival. Moreover, Morgana/NF-κB axis is responsible for cytokine production by cancer cells, shaping the composition of the immune-cell infiltrate in the primary tumor and in the lung pre-metastatic niche.

## Results

### Morgana is essential for TNBC cell invasion and metastasis.

We recently demonstrated that Morgana is frequently over-expressed in TNBC and its expression correlates with lymph node positivity[14]. Accordingly, Morgana is expressed at higher levels in MDA-MB-231 and BT549 TNBC cell lines compared to non-invasive MCF7 cancer cells and MCF10A normal breast cells (Fig. 1a). We silenced Morgana in MDA-MB-231 and BT549 cell lines using two different shRNAs (Fig. 1b and Supplementary Fig. 1a). In both cell lines Morgana downregulation did not cause differences in in vitro proliferation (Fig. 1c and Supplementary Fig. 1b).

One of the first steps in cancer cell metastasis is the acquisition of invasive properties. MDA-MB-231 silenced for Morgana displayed an impaired invasive capacity in Matrigel-coated transwell assays if compared with control cells (Fig. 1d).

Morgana silencing significantly affected the anchorage-independent growth ability of these cells in a soft agar assay, another feature of metastatic cells (Fig. 1e). These data suggested an involvement of Morgana in different steps of cancer cell metastasis.

To validate these findings in vivo, MDA-MB-231 cells were injected in the tail vein of NOD/SCID/IL-2Rγc null (NSG) mice. After 4 weeks all the mice receiving MDA-MB-231 infected with empty vector exhibited macrometastases and numerous micrometastases in lungs, liver, kidneys, and heart[25, 26]; while only few and small micrometastases were found in the lungs of mice receiving MDA-MB-231 infected with two independent shRNAs targeting Morgana (Fig. 1f–i).

To investigate the role of Morgana expression in cancer cell metastasis from primary tumors, NSG mice were inoculated subcutaneously with MDA-MB-231 expressing two shRNAs against Morgana or infected with an empty vector. While no differences were detected in primary tumor growth[14] (Fig. 1j), after 5 weeks MDA-MB-231 infected with the empty vector formed macrometastases in the lungs, conversely cells downregulated for Morgana did not form any macrometastases and only few micrometastases were detected (Fig. 1k–m).

In the TNBC cell line BT549, unable to form metastasis in vivo, the role of Morgana in invasion has been confirmed through its downregulation and overexpression followed by in vitro invasion assays (Supplementary Fig. 1c). Strikingly, BT549 overexpressing Morgana acquired the ability to form metastases in the lung in a spontaneous metastasis assay in NSG mice (Supplementary Fig. 1d–g).

We previously demonstrated that Morgana binds to and inhibits ROCK I and II[14, 20] and that through ROCK I inhibition Morgana downregulates PTEN expression levels thus enhancing AKT phosphorylation[14]. Rho kinases play a pivotal role in adhesion and cytoskeletal regulation and although their fine tuning is required to allow cell movement[27], high ROCK activity is generally associated to an increase in cancer cell invasion[28–32]. In this view, Morgana downregulation, leading to ROCK activation, would be expected to boost cancer cell motility, rather

**Fig. 1** Morgana is essential for TNBC cell invasion and in vivo metastasis formation. **a** Immunoblotting of Morgana and Vinculin, as loading control, in MCF7, MDA-MB-231, BT549 and MCF10A cell lines. **b** Immunoblotting of Morgana and Vinculin in MDA-MB-231 cells infected with empty vector (EMPTY) or two independent Morgana shRNAs (shMORG1, shMORG2). **c** Growth curves of MDA-MB-231. **d** Quantification of invasion assays performed on MDA-MB-231 EMPTY or shMORG1 and shMORG2. **e** Quantification of soft agar colony formation by MDA-MB-231 EMPTY or shMORG1 and shMORG2. **f–h** Representative haematoxylin and eosin stained sections **f**, quantification of **g** pulmonary metastases and **h** metastatic burden, 4 weeks after tail vein injection of MDA-MB-231 EMPTY or shMORG1 and shMORG2 (n = 6 NSG mice per group). **i** Metastasis quantification in distant organs in mice injected with MDA-MB-231 EMPTY or shMORG1 and shMORG2 (n = 3 NSG mice per group). **j** Tumor weight 5 weeks after subcutaneous injection of MDA-MB-231 EMPTY or shMORG1 and shMORG2 (n = 6 NSG mice per group). **k–m** Representative haematoxylin and eosin stained sections **k**, quantification of **l** pulmonary metastases and **m** metastatic burden of MDA-MB-231 EMPTY or shMORG1 and shMORG2 (n = 6 NSG mice per group). Data are the results of at least three independent experiments. Bars in graphs represent standard errors (*p < 0.05; **p < 0.01; ***p < 0.001)

than inhibiting it. However, we tested the involvement of ROCK inhibition in Morgana induced invasion using an in vitro matrigel transwell assay. The ROCK inhibitor Y27632 was unable to rescue the ability to invade of Morgana deficient cells (Supplementary Fig. 2a, b), suggesting the involvement of a different pathway.

**Morgana silencing in TNBC cells reduces MMP9 expression**. Degradation of extracellular matrix and vascular basement membrane is required for a cancer cell to invade and form

metastasis and it is known that matrix metalloproteinases, in particular MMP9, are essential for invasion of TNBC cells[33]. In the attempt to find clues on the mechanism by which Morgana induces metastasis formation, we analyzed if metalloproteinase activity could be dependent on Morgana. Zymography assays indicated that both extracellular and intracellular MMP9 activity were strongly impaired in MDA-MB-231 cells silenced for Morgana compared with control cells (Fig. 2a, b). Indeed, Western blot analysis of extra-cellular and intra-cellular MMP9

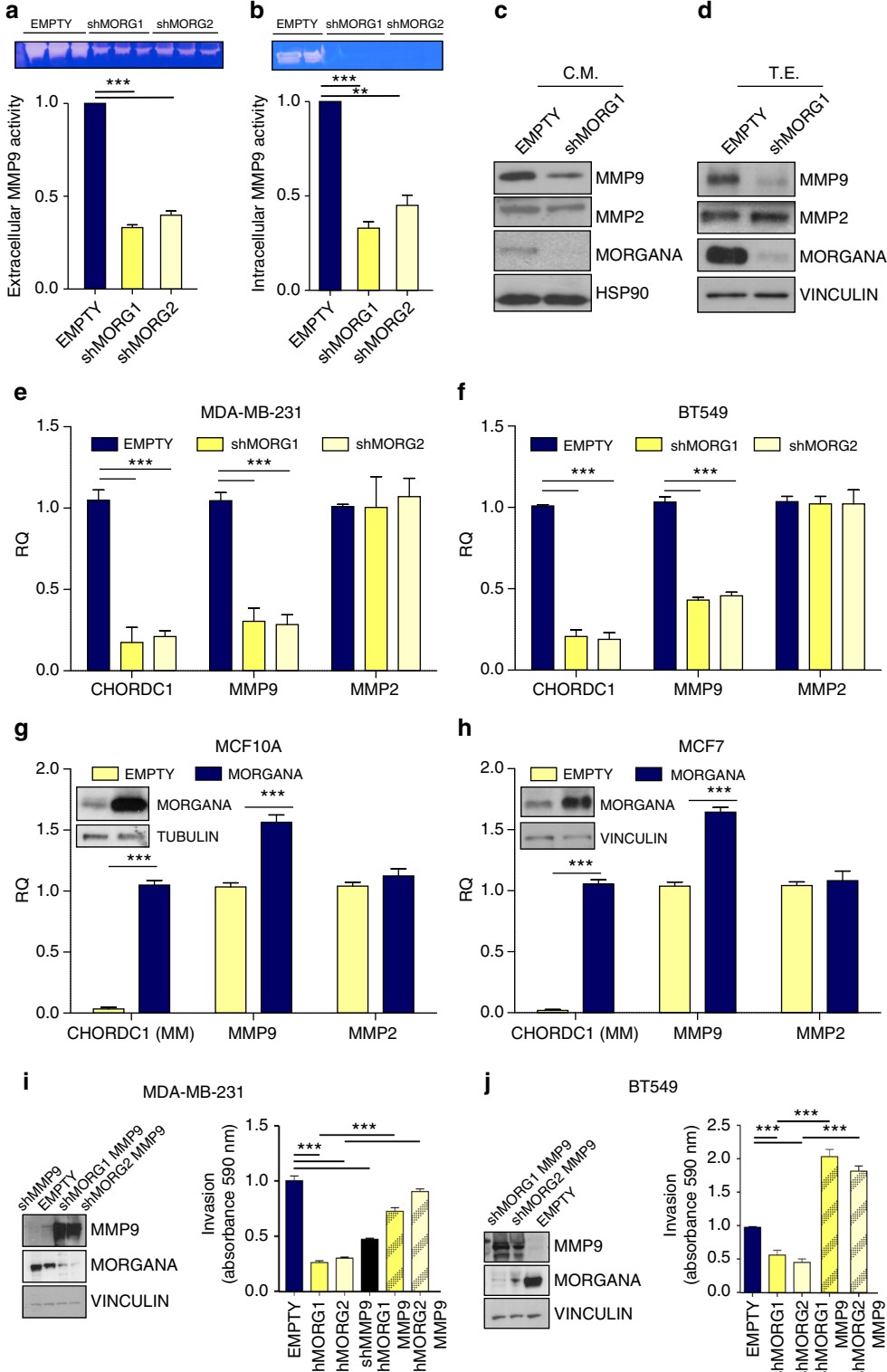

revealed a relevant decrease in MMP9 expression in Morgana downregulated cells (Fig. 2c, d). Real-time PCR analysis demonstrated that Morgana downregulation affects specifically MMP9 mRNA levels since no differences were observed for MMP2 (Fig. 2e, f). MCF10A and MCF7 cells express low Morgana levels if compared with MDA-MB-231 (Fig. 1a)[14] and a very low level of MMP9 mRNA[34]. Morgana overexpression in MCF10A and MCF7 cells induced a significant increase in MMP9 transcription (Fig. 2g, h). Of note, MMP9 overexpression rescued invasion abilities both in MDA-MB-231 and BT549 down-regulated for Morgana (Fig. 2i, j). Given that PI3K/AKT pathway has been reported to induce MMP9 transcription[35], we tested the involvement of ROCK and PTEN in MMP9 expression in our system. ROCK and PTEN inhibition was unable to rescue MMP9 expression in MDA-MB-231 silenced for Morgana (Supplementary Fig. 2c–e), again suggesting that Morgana acts through a different pathway in this context.

**NF-κB is responsible for Morgana dependent metastasis.** NF-κB and AP-1 are the most important transcription factors known to induce MMP9. To evaluate if Morgana can influence their activity we used a luciferase reporter gene with specific AP-1 or NF-κB binding sites in the promoter region. Morgana down-regulation significantly decreased NF-κB, but not AP-1 tran-scriptional activity, both in MDA-MB-231 and BT549 (Fig. 3a and Supplementary Fig. 3a) and its overexpression in MCF10A and MCF7 cells induced a specific increase of NF-κB dependent luciferase activity, independently from PI3K and AKT activation (Supplementary Fig. 3b–f and Fig. 3b). To further extend our analysis we used immortalized cell lines (NIH-3T3 and HEK293, in which Morgana has been downregulated or overexpressed, and primary mouse embryonic fibroblasts (MEFs) from morgana+/-mice[20] and wild type controls. In all these cellular models a higher Morgana expression caused an increase in NF-κB transcriptional activity (Supplementary Fig. 3g–i). In accordance, in both MDA-MB-231 and BT549, Morgana silencing induced a significant reduction in the expression of several NF-κB target genes (Fig. 3c and Supplementary Fig. 4a) and conversely, Morgana over-expression in MCF7 and MCF10A caused an increase in their expression levels (Fig. 3d and Supplementary Fig. 4b). Further, alteration of Morgana expression levels in HeLa, HEK293, NIH-3T3 and MEFs caused a parallel variation in NF-κB target gene expression (Supplementary Fig. 4c–f). Gene Ontology enrichment on data obtained in a microarray analysis (GEO ID: GSE86463), indicated that genes down-regulated in MDA-MB-231 silenced for Morgana vs. control cells are mainly involved in inflammation and immune system regulation (Fig. 3e and Supplementary Table 1). GSEA analysis showed that NF-κB target gene sig-natures are globally downregulated in silenced cells (Supplementary Table 2).

To determine if Morgana dependent metastasis formation in vivo was due to NF-κB activation, we expressed an IκBα shRNA or an empty vector in MDA-MB-231 in which Morgana was downregulated (Fig. 3f). As expected, IκBα silencing induced NF-κB activation as demonstrated by a luciferase assay (Fig. 3g). Notably, IκBα depletion completely reverted the metastatic impairment due to Morgana downregulation in MDA-MB-231 both in experimental (Fig. 3h, i) and spontaneous metastasis assays (Fig. 3j–l).

**Morgana is an essential component of the IKK complex.** Next, we used TNBC cell lines, in which NF-κB is constitutively hyperactive, and MCF7 and MCF10A, in which NF-κB is barely activated in absence of stimuli[12], to analyze how Morgana can impact on NF-κB upstream signaling pathway. IκBα is phos-phorylated on serines 32 and 36 by the IKK complex. IκBα phosphorylation on these residues was robustly decreased in MDA-MB-231 and BT549 cells silenced for Morgana (Fig. 4a, b and Supplementary Fig. 5a, b), while no differences were observed in the expression of several components of the NF-κB canonical pathway (Supplementary Fig. 5a, b). Conversely, in both MCF10A and MCF7, Morgana overexpression increased IκBα phosphorylation specifically on serines 32 and 36 (Fig. 4c, d). Alteration in Morgana expression levels caused a corresponding variation in IκBα phosphorylation also in other cell types including HeLa, HEK293, NIH-3T3 and MEFs (Supplementary Fig. 5c–f). Immunoprecipitation experiments demonstrated that Morgana interacts with IκBα and the IKK complex both in MDA-MB-231 and BT549 (Fig. 4e, f) and in non-tumorigenic cell lines HEK293 and NIH-3T3 (Supplementary Fig. 6a, b). Further, Morgana directly bound to IκBα in a Far Western experiment (Supplementary Fig. 6c). In accordance, by immunoprecipitating IKKβ or IKKα, Morgana was found in the complex (Supple-mentary Fig. 6d, e). In gel filtration analysis of MDA-MB-231 total extracts, Morgana eluted together with IKKα and IKKβ in fractions around 1000–900 kDa (Fig. 4g). Co-immunoprecipitation experiments from these fractions demon-strated that Morgana binds to the mature IKK complex[36] (Fig. 4h). To evaluate the possibility that Morgana plays a func-tion in the IKK complex assembly, we compared gel filtration experiments on total extracts from Morgana depleted and control cells. As shown in Fig. 4i, IKKα and IKKβ subunits eluted in the same high-molecular fractions, indicating that IKK complex assembly was not affected by Morgana silencing. It is to note that the Morgana interactor Hsp90[17–22], which is required for IKKγ recruitment to the complex[36], eluted together with IKK high molecular complex also in Morgana downregulated cells (Fig. 4i). Moreover, silencing of Hsp90 and its co-chaperone cdc37 did not impair Morgana binding to IκBα and IKK complex (Supple-mentary Fig. 6f, g). Instead, Morgana is essential for IκBα recruitment to the IKK complex, as assessed by IKKβ immuno-precipitation from MDA-MB-231 (Fig. 4j). In an in vitro IKKβ kinase assay the phosphorylation of a recombinant IκBα substrate was significantly impaired in Morgana downregulated cells

**Fig. 2** Morgana silencing in TNBC cells reduces MMP9 expression. **a**, **b** Extracellular **a** and intracellular **b** MMP9 activity in MDA-MB-231 infected with an empty vector (EMPTY) or shRNAs targeting Morgana (shMORG1 and shMORG2) was evaluated by gelatin zymography. Top panel: representative zymogram of MMP9 activity (92 kDa). Bottom panel: quantification of MMP9 activity from 3 independent experiments. **c** Western blot analysis of conditioned medium (C.M.) obtained from MDA-MB-231 EMPTY or shMORG1 immunostained with MMP9, MMP2, Morgana and extracellular HSP90 as loading control. Morgana, as other chaperone proteins, can be secreted by cancer cells. **d** Western blot analysis of total protein extracts (T.E.) from MDA-MB-231 EMPTY and shMORG1 immunostained for MMP9, MMP2, Morgana and Vinculin as loading control. **e**, **f** Gene expression analysis by Real-time PCR of MMP9, MMP2 and CHORDC1 (Morgana coding gene) in MDA-MB-231 **e** and BT549 **f** EMPTY or shMORG1 and shMORG2. **g**, **h** Gene expression analysis by Real-time PCR of MMP9, MMP2 and CHORDC1 (Morgana coding gene) in MCF10A **g** and MCF7 **h** overexpressing Morgana (MORGANA) or control cells (EMPTY). **i** Quantification of invasion assays performed on MDA-MB-231 EMPTY or shMORG1 and shMORG2 infected or not with a lentiviral vector coding for MMP9 (shMORG1 MMP9, shMORG2 MMP9) or a shRNA targeting MMP9 (shMMP9). **j** Quantification of invasion assays performed on BT549 EMPTY or shMORG1 and shMORG2 infected or not with a lentiviral vector coding for MMP9 (shMORG1 MMP9, shMORG2 MMP9). Data are the results of at least three independent experiments. Bars in graphs represent standard errors (**$p < 0.01$; ***$p < 0.001$)

compared to controls (Fig. 4k). However, the addition of an MBP-Morgana recombinant protein to the immunoprecipitate restored IκBα phosphorylation, while the recombinant MBP alone had no effect (Fig. 4k). This data indicates that Morgana is required per se to induce IκBα phosphorylation by IKKβ, without the need to recruit further elements to the complex.

To understand whether Morgana activates IKKβ or mediates the recruitment of IκBα to the IKK complex we transfected MDA-MB-231 and BT549 silenced for Morgana and control cells with a wild type or a constitutively active IKKβ. Our results showed that even in cell transfected with the constitutively active IKKβ, the IκBα phosphorylation is totally impaired when

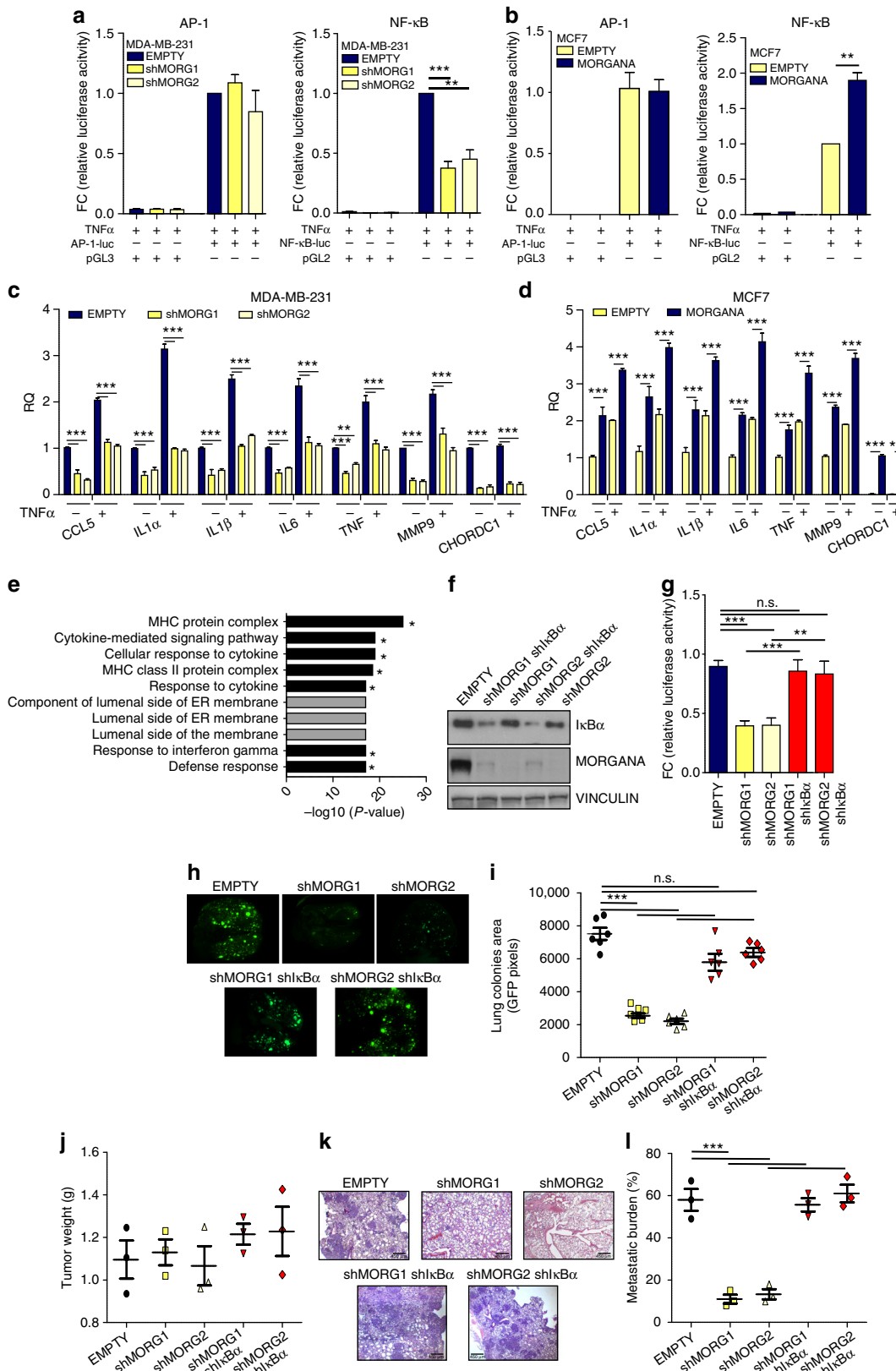

Morgana is silenced (Fig. 4l and Supplementary Fig. 6h). Taken together, these results indicate a role for Morgana in mediating the binding of the IKK complex to its substrate.

**Morgana is co-expressed with NF-κB target genes.** To evaluate the relevance of our molecular results in human breast cancer, we used a cohort of 152 patients from San Giovanni Battista Hospital (SGBH cohort) and bioinformatics analysis on TCGA datasets. This approach is warranted by the presence of a positive correlation between Morgana expression assessed by immunohistochemistry[14] and by Real-time PCR in breast cancer tissue array specimens ($n = 58$) (Supplementary Fig. 7a).

Paired expression data for breast tumors and adjacent normal tissue were available for 113 samples in TCGA dataset. Morgana was significantly upregulated in tumors compared to adjacent normal tissue (Supplementary Fig. 7b) and its expression differed significantly among molecular subtypes being more expressed in basal-like and less expressed in luminal A and B subtypes (Supplementary Fig. 7c, d). Notably, Morgana expression significantly correlated with poor survival both in the TCGA dataset (Fig. 5a) and in the SGBH cohort of patients (Fig. 5b). In keeping with our molecular results, analyzing the TCGA protein array data in breast cancer specimens, we identified a significant correlation between Morgana expression and NF-κB activity, assessed by the analysis of p65 phosphorylation on serine 536[37] (Fig. 5c). Moreover, GSEA analysis showed that several NF-κB gene signatures display positive correlation with Morgana expression in the breast cancer TCGA dataset (Fig. 5d, e). In particular, the expression of two well-known NF-κB target genes, IL-1β and MMP9, correlated with Morgana expression (Fig. 5f, g). Furthermore, analyzing by Real-time PCR 58 breast cancer samples from our SGBH patient cohort, we found a significant correlation between Morgana and MMP9 expression levels (Fig. 5h). In accordance, on this cohort, the invasion grade was significantly higher in patients expressing high Morgana levels (Fig. 5i).

Moreover, gene set enrichment analysis on a panel of breast cancer cell lines[38] showed that in the whole panel as well as in the TNBC cell line subset, Morgana expression levels positively correlated with the NF-κB gene signature (Supplementary Fig. 7e, f). Of note, Morgana expression analysis on the 22 TCGA tumors, for which data on adjacent normal tissue were present, indicated that Morgana is significantly upregulated in 40% (9 out of 22) and downregulated in 4% (1 out of 22) tumor types compared to normal tissue (Supplementary Fig. 7g, h).

**Morgana modulates immune cell recruitment.** To evaluate the role of Morgana in orchestrating pathological immune system engagement in cancer progression, we used two transplantable models in syngeneic immunocompetent mice. As in human breast cancer cells, Morgana downregulation in 4T1 and E0771

mouse mammary carcinoma cells (Supplementary Fig. 8a, b), whereas not affecting proliferation in vitro (Supplementary Fig. 8c, d), clearly impacted on IκBα phosphorylation (Supplementary Fig. 8e, f) and NF-κB target gene transcription (Supplementary Fig. 8g, h).

To evaluate if Morgana expression in cancer cells engraves on the microenvironment ability to sustain carcinogenesis, 4T1 and E0771 silenced for Morgana or control cells were injected subcutaneously respectively in BALB/c and C57BL/6 syngeneic female mice. Differently from in vitro culture (Supplementary Fig. 8c, d), 4T1 and E0771 cancer cells silenced for Morgana grew significantly less in mice (Fig. 6a and Supplementary Fig. 9a). Moreover, the ability of Morgana to regulate NF-κB activity was retained in vivo, since the transcription of different NF-κB target genes in the primary tumor was reduced when Morgana expression was downregulated (Supplementary Fig. 9b, c). Cytofluorimetric analysis of disaggregated primary tumors (30 days from cell injection) indicated that Morgana depleted tumors recruited less neutrophils (Fig. 6b, c and Supplementary Fig. 9d, e). Both primary tumor growth and neutrophil recruitment were totally rescued by IκBα silencing in 4T1 and E0771 downregulated for Morgana (Fig. 6a–e and Supplementary Fig. 9a, d–g), demonstrating the causative role of NF-κB in these events. Analysis at an early time point (10 and 14 days from 4T1 and E0771 injection, respectively), when tumor sizes were still comparable, confirmed that neutrophil recruitment in primary tumor depends on Morgana expression (Fig. 6f–h and Supplementary Fig. 10a–c). Even earlier (4 and 6 days from injection of 4T1 and E0771), we noticed a significant increase in natural killer (NK) cells in Morgana downregulated tumors (Fig. 7a–c and Supplementary Fig. 10d–f). No differences were found in other immune cells (Figs. 6b, g and 7b, Supplementary Figs. 9d, 10b, e, 11a, b) and endothelial cells at all time points (Supplementary Fig. 11c, d). NK cells are innate lymphoid cells playing a relevant role in cancer immune surveillance. Their cytotoxic activity is regulated by a number of activating and inhibitory signals triggered by cell surface receptors on the target cell[39]. Among others, the expression of MHC class I receptors on cancer cells potently inhibits NK activation. Accordingly, microarray and real time PCR analysis on MDA-MB-231, 4T1 and E0771 interfered for Morgana (Fig. 7d–f) highlighted a decrease in MHC class I receptors. Of note, MDA-MB-231 and BT549 silenced for Morgana did not form tumor in nude mice, where, unlike in NSG mice, NK cells are present and functional[26] (Fig. 7g, h). Besides differences in primary tumor growth, we observed a dramatic impairment in spontaneous lung metastasis formation in syngeneic mice carrying Morgana downregulated tumors that it was rescued by IκBα silencing (Fig. 8a, b). Recently, it has been demonstrated that neutrophils, other than promoting tumor growth and progression[40], are crucial in preparing the breast cancer pre-metastatic niche in the lung[41, 42]. Very interestingly, neutrophil recruitment in mice carrying early stage tumors, when

**Fig. 3** NF-κB pathway is responsible for Morgana dependent cancer cell metastasis. **a**, **b** Luciferase assays of AP-1 (left) and NF-κB (right) activity in **a** MDA-MB-231 infected with an empty vector (EMPTY) or shRNAs targeting Morgana (shMORG1 and shMORG2) and **b** MCF7 infected with an empty vector (EMPTY) or overexpressing Morgana (MORGANA). Cells were transfected with an AP-1 or NF-κB luciferase reporter or a control vector. **c**, **d** Gene expression analysis by Real-time PCR of NF-κB target genes in **c** MDA-MB-231 EMPTY or shMORG1 and shMORG2, and **d** MCF7 EMPTY or MORGANA, treated or not with 10nM TNFα for 4 h. **e** The top 10 gene ontology (GO) terms by enrichment P-value among the down-regulated genes obtained in a microarray analysis of MDA-MB-231 cells silenced for Morgana compared to control cells. The stars indicate signatures NF-κB related. **f** Immunoblotting of IκBα, Morgana and Vinculin in MDA-MB-231 cells EMPTY or shMORG1 and shMORG2 in combination or not with IκBα shRNA. **g** Luciferase assays of NF-κB activity in MDA-MB-231 cells described in **f** transfected with a NF-κB luciferase reporter or control vector. **h** Representative pictures of lungs of NSG mice after 4 weeks from the intravenous injection of MDA-MB-231 cells described in **f**. **i** Quantification of metastasis colonies in the lungs of mice described in **h** ($n = 6$ NSG mice per group). **j** Tumor weight 5 weeks after subcutaneous injection of cells described in **f** ($n = 3$ NSG mice per group). **k**, **l** Representative haematoxylin and eosin stained sections **k** and percentage of lung metastatic area **l**. Each data point represents one mouse ($n = 3$ NSG mice per group). Data are the results of three independent experiments. Bars in graphs represent standard errors (**p < 0.01; ***p < 0.001)

no differences in size were present, was heavily reduced in the lungs of mice injected with Morgana silenced cells (Fig. 8c–f). Accordingly, the expression in the lungs of neutrophil-attracting chemokines and pre-metastatic niche markers was reduced (Fig. 8g). To demonstrate the relevance of Morgana expression in recruiting immune cells in human tumors, we performed a

correlation between Morgana expression levels and genes typically expressed by neutrophils, lymphocytes and macrophages using the TCGA data set expression data. Morgana expression levels positively correlated with neutrophil markers FCGR3A (CD16) and FCGR3B (CD16b) (Fig. 8h), while no correlation was detected with lymphocyte (MS4A1 coding for CD20 and CD19)

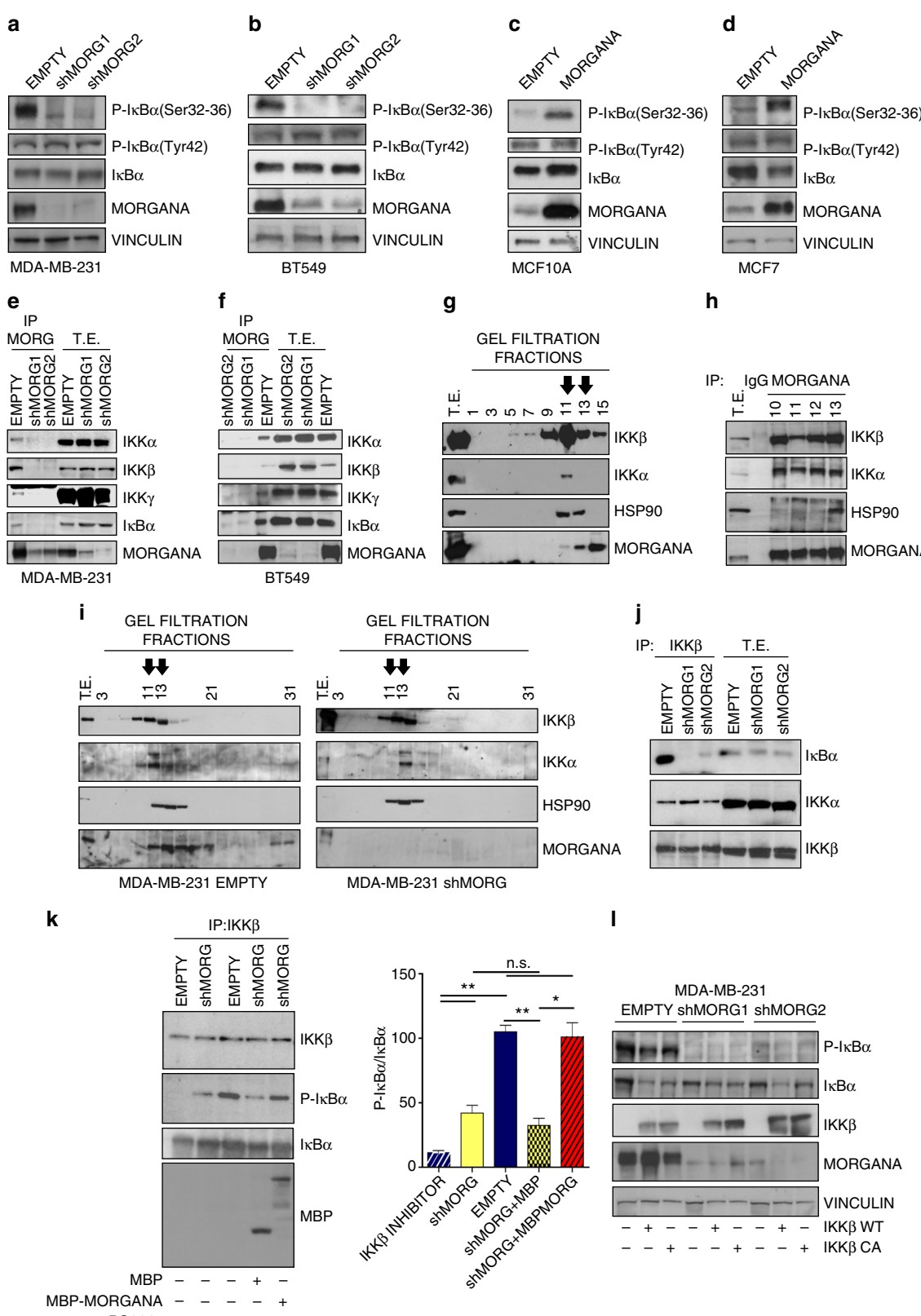

and macrophage markers (CD68 and CD14) (Supplementary Table 3), indicating a specific role of the Morgana/NF-κB axis in neutrophil recruitment also in human cancers.

Overall these results indicate that Morgana expression in tumor cells is necessary to produce cancer cell immune escape and neutrophil recruitment in primary tumor and pre-metastatic niche (Fig. 9).

## Discussion

The IKK complex is composed by two serine-threonine kinases (IKKα and IKKβ) and by the regulatory scaffold protein IKKγ/NEMO. It is known that phosphorylation of IKKα and IKKβ and the interaction of IKKγ/NEMO with K63 poly-ubiquitinilated chains[43] are required for activation of the complex. Considering the relevance of the role played by NF-κB in cancer, dissecting the precise mechanism by which the IKK complex integrates different signals, gets activated and recognizes its substrates is fundamental. Although tremendous progress has recently been made in resolving the structure of the IKK complex components, it is still unclear if different IKK complexes exist in the cells and which their exact compositions and specific functions are[5, 6]. Nevertheless, different IKK interactors have been described[44–46]. Among them the protein ELKS has been characterized as an IKK regulatory subunit, recruiting IκBα to the IKK complex in HeLa cells[47]. Interestingly, the chaperone proteins Hsp70, Hsp90 and its co-chaperone cdc37 also bind to the IKK complex. While Hsp70 behaves as an IKKγ inhibitor, Hsp90 and cdc37 stabilize the IKK complex and are required for its activation[36, 48–50]. In this work we describe Morgana as a previously unknown component of the IKK complex required for NF-κB downstream activation in breast cancer cells. Morgana, besides its ability to inhibit Rho kinases[14, 20], displays chaperone activity per se and behaves as a Hsp90 co-chaperone[17–19, 22]. It is worth noting that Morgana does not impact on IKKα, β, and γ stability and expression levels. We showed that Morgana interacts with the mature high molecular weight IKK complex, but it is not required for IKK complex formation. However, Morgana is essential for the recruitment of IκBα to the IKK complex and ultimately for its phosphorylation. Morgana is sufficient to accomplish this task, since the addition of a Morgana recombinant protein to IKKβ immunoprecipitate from Morgana depleted cells restores IκBα phosphorylation in an in vitro kinase assay. In line with our molecular data, Morgana potently regulates NF-κB transcriptional activity and target gene expression in different breast cancer cell lines. This also happens in non-tumorigenic and primary cells, indicating that Morgana may act as a general regulator of NF-κB activity. However, further work is needed to understand the relevance of Morgana in other cellular contexts and in response to different NF-κB activating stimuli. Importantly, Morgana silencing, by inhibiting NF-κB pathway, was able

to almost completely block cancer cell metastasis. Bioinformatics studies on TCGA breast cancer dataset, coupled with analysis on a cohort of 152 breast cancer patients, confirmed a correlation between Morgana expression levels, NF-κB activity and poor survival in human breast cancer patients.

Cancer cells engage essential relationship with normal cells creating a favorable environment for its survival and growth. Specifically, they recruit mesenchymal and inflammatory cells from the circulation into the tumor microenvironment and induce their pathological activation by secreting a plethora of cytokines. In turn, inflammatory cells and activated fibroblasts support cancer growth and progression producing cytokines and soluble factors. This cancer-microenvironment interaction leads to the establishment of a vicious circle capable of maintaining a chronic inflammatory state and fuel cancer growth and progression[51, 52]. In the mean time, cancer cells must evade immune surveillance, through a number of mechanisms such as targeting the regulatory T cell function or inducing immune system tolerance or ignorance. Our data indicate that Morgana signaling plays a fundamental role both in inducing immune escape and in recruiting ancillary cells in the tumor microenvironment. Indeed, high Morgana expression level in cancer cells inhibits the recruitment of NK cells during the very early phases of cancer formation, while it induces neutrophil accumulation in subsequent steps. The ability of NK cells to recognize and kill cancer cells depends on the balance between activating and inhibitory signals. MHC class I receptors on cancer cells bind to NK receptors and potently inhibit their activation[39]. We discovered that the expression of MHC class I receptors on cancer cells depends on Morgana, likely being regulated by NF-κB activity[53]. Moreover, the fact that human breast cancer cells silenced for Morgana grow in NSG mice but not in nude mice, where NK cells are present and functional[26], further supports a role for Morgana in cancer cell immune escape through NK inhibition. Considering the innovative therapeutic approaches to activate NK anti-tumor activity, Morgana overexpression may represent an important biomarker to direct clinical intervention towards the blockage of NK inhibitory receptors[39, 52].

The pivotal role of neutrophils in cancer onset and progression has been recently recognized[40, 54, 55]. Cancer-derived chemokines induce neutrophil proliferation and egression from the bone marrow and their polarization toward a pro-tumor phenotype, able to promote primary tumor growth and spread[40]. NF-κB activation, through IκBα silencing, in cancer cells downregulated for Morgana totally rescues primary tumor growth and neutrophil recruitment, demonstrating a causal role of Morgana/NF-κB signaling in these events. This is in line with previous results showing that IL-1β, a well known NF-κB target gene, is the first player triggering a systemic signaling responsible for neutrophil expansion and polarization in a mouse model of spontaneous

**Fig. 4** Morgana is an essential component of the IKK complex in cancer cells. **a**, **b** Western blot of MDA-MB-231 **a** and BT549 **b** infected with an empty vector (EMPTY) or two Morgana shRNAs (shMORG1, shMORG2). **c**, **d** Immunoblot of MCF10A **c** and MCF7 **d** infected with an empty vector (EMPTY) or overexpressing Morgana (MORGANA). **e**, **f** Immunoprecipitation of Morgana from MDA-MB-231 **e** or BT549 **f** immunoblotted with IKKα, IKKβ, IKKγ, IκBα and Morgana. **g** Gel filtration analysis of the endogenous IKK complex purified from MDA-MB-231. The total protein extracts were fractionated on a Superose 6 gel filtration column. Each fraction was analyzed by Western blot. The arrows indicate the fractions in which Morgana and the IKK complex eluted together. **h** Immunoprecipitation of Morgana from selected fractions obtained by gel filtration immunochromatography of MDA-MB-231 immunoblotted with IKKα, IKKβ, HSP90 and Morgana. **i** Gel filtration analysis of total protein extracts from MDA-MB-231 shMORG1 (right) compared with control cells (left). Each fraction was analyzed by Western blot. The arrows indicate the fractions in which Morgana and the IKK complex eluted together. **j** Immunoprecipitation of IKKβ from MDA-MB-231 immunoblotted with IκBα, IKKα and IKKβ. **k** IKKβ was immunoprecipitated from MDA-MB-231 EMPTY and shMORG and subjected to an in vitro kinase assay with or without addiction of recombinant MBP-MORGANA or MBP as control. The IKKβ inhibitor, PS1145, was added to IKKβ immunoprecipitation from MDA-MB-231 EMPTY, as control. The graph shows the average intensity of the P-IκBα bands normalized to IκBα. **l** Immunoblotting of P-IκBα, IκBα, IKKβ, Morgana and Vinculin in MDA-MB-231 EMPTY or shMORG1 and shMORG2 transfected or not with IKKβ wild type (WT) or constitutively active (CA). Data are the results of three independent experiments. Bars in graphs represent standard errors (*p < 0.05; **p < 0.01)

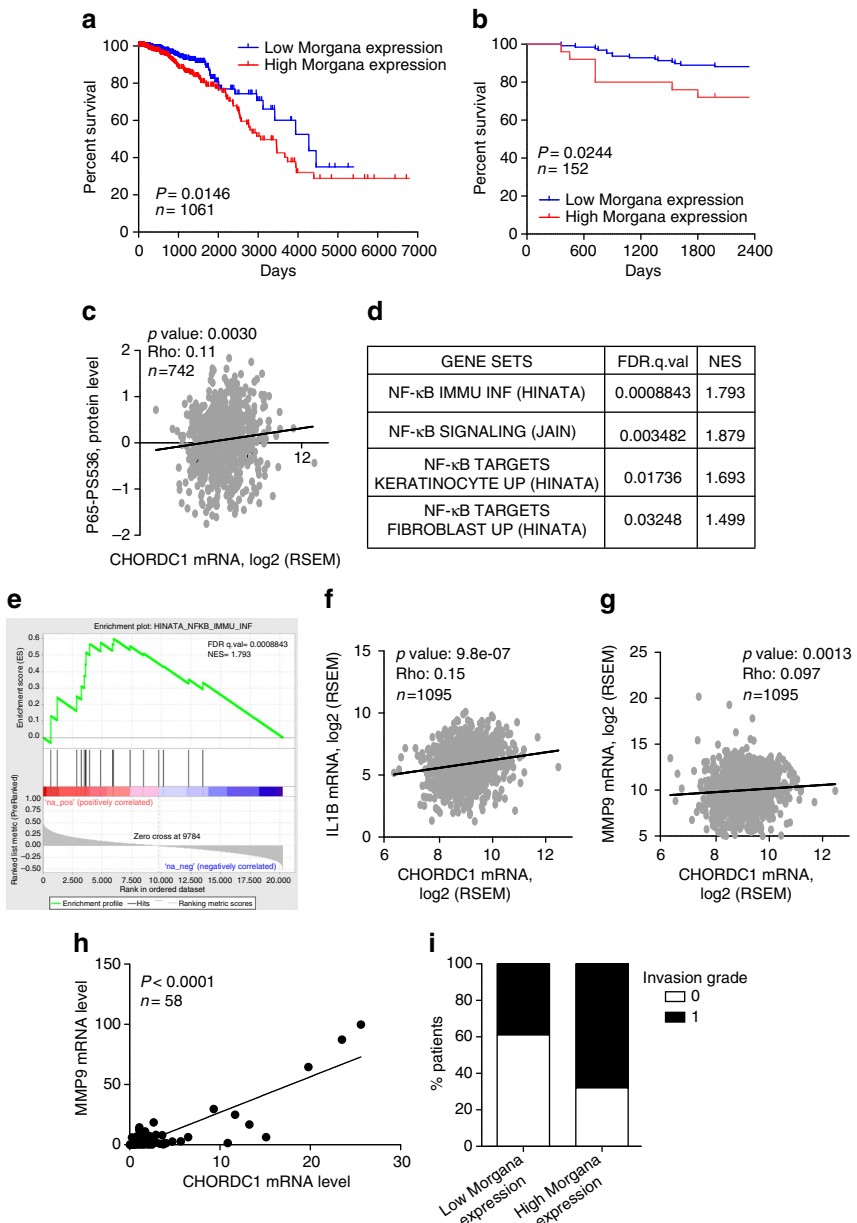

**Fig. 5** Morgana is co-expressed with NF-κB target genes in human breast cancers. **a** Morgana expression significantly correlated with poor survival in TCGA dataset ($P = 0.0146$, Mantel-Cox test). **b** Morgana expression significantly correlated with poor survival in San Giovanni Battista Hospital (SGBH) cohort of 152 patients ($P = 0.0244$, Mantel-Cox test). **c** Spearman correlation between CHORDC1 (Morgana coding gene) mRNA levels and the phosphorylation status of NF-κB p65 in Ser536 in 742 samples of TCGA dataset. **d** Table showing signature positively correlated with CHORDC1 mRNA levels generated by GSEA analysis of ranked NF-κB gene expression data in TCGA dataset. **e** Gene set enrichment analysis plot of the most CHORDC1-positively correlated NF-κB dataset. **f, g** Spearman correlation in mRNA levels between CHORDC1 and IL1B **f** and MMP9 **g** in TCGA data set ($n = 1095$). **h** Spearman correlation in mRNA levels between CHORDC1 and MMP9 in 58 patients from SGBH cohort. **i** Correlation between Morgana expression levels and vascular invasion grade indicated as 0 or 1 in 152 human breast cancer samples of SGBH cohort

metastasis[42]. Besides being recruited to the primary tumor, neutrophils accumulate in the pre-metastatic niche and drive metastatic initiation[41, 42, 56]. Morgana downregulation in breast cancer cells, by quenching NF-κB signaling, inhibits neutrophil recruitment and the formation of the pre-metastatic niche in the lung, preventing metastasis formation.

In this work we identified Morgana as a component of the IKK complex required for IκBα phosphorylation and for NF-κB activation, adding a piece to the complex NF-κB puzzle[57]. Moreover, our data indicate that increase in Morgana expression levels may represent an opportunity for cancer cells to evade immune surveillance and to amplify their ability to shape the

microenvironment enhancing chemokine production, and, at the same time, to acquire hypersensitivity to inflammatory stimuli generated by infiltrating cells, strengthening the aberrant symbiosis between cancer and the immune system. Finally, Morgana/NF-κB axis in the primary tumor is responsible for changing in systemic environment required for pre-metastatic niche generation and cancer progression.

## Methods

**Cell culture.** The human and murine cell lines MDA-MB-231, HEK293, mouse embryonic fibroblasts (MEFs) and 4T1 were cultured in DMEM medium (Thermo Fisher Scientific) with 10% FBS (Euroclone) and 1% Penicillin Streptomycin

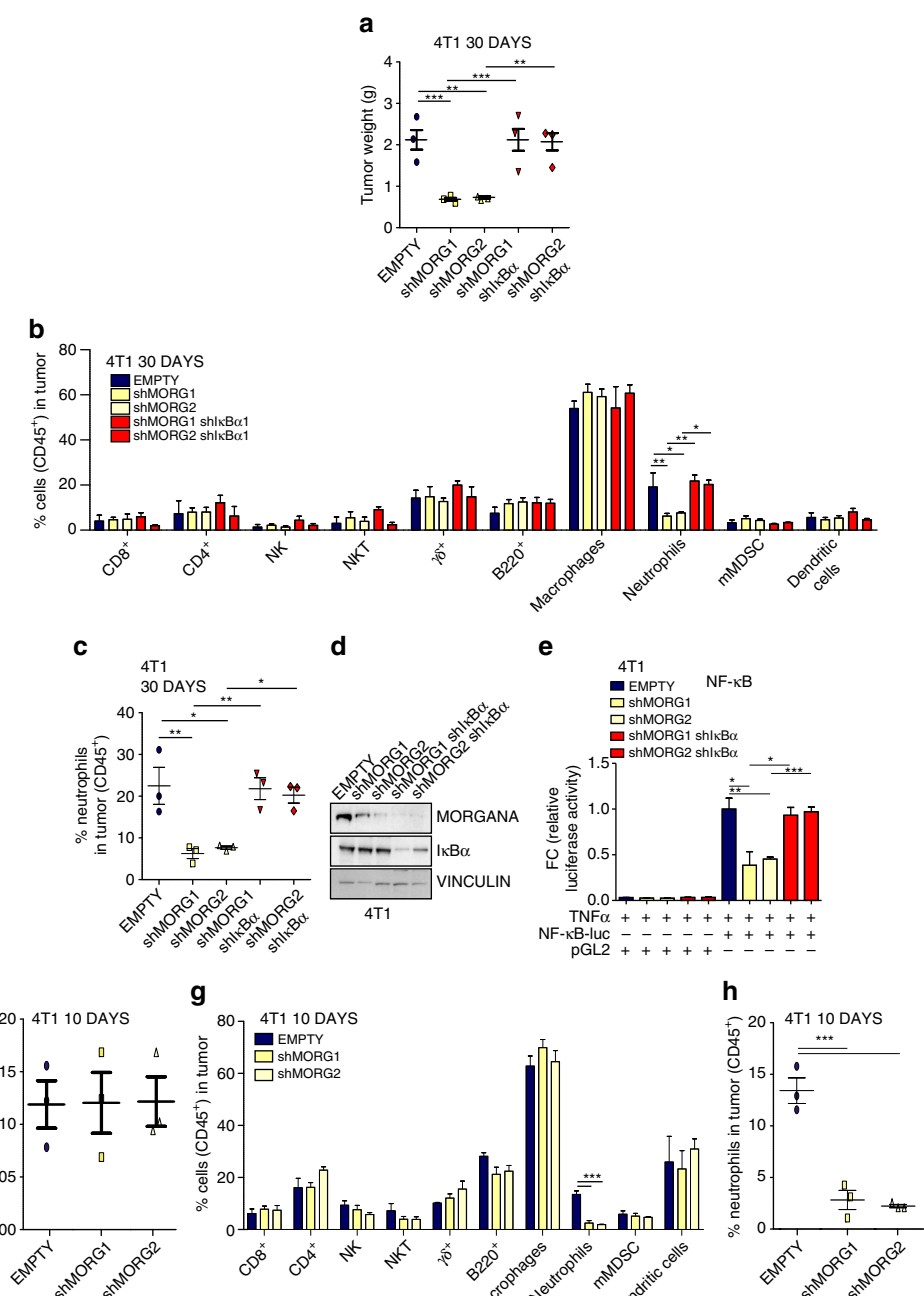

**Fig. 6** Morgana-NF-κB axis in breast cancer cells induces neutrophil recruitment in primary tumor. **a** Tumor weight 30 days after subcutaneous injection of 4T1 cells EMPTY or shMORG1 and shMORG2 infected or not with IκBα shRNA ($n = 3$ BALB/c mice per group). **b** Percentage of immune cells (gated on CD45$^+$ cells) detected in the primary tumor 30 days after injection of 4T1 described in **a** ($n = 3$ BALB/c mice per group). **c** Percentage of neutrophils detected in the primary tumor of each mouse 30 days after injection of 4T1 cells described in **a**. **d** Immunoblotting of Morgana, IκBα and Vinculin in 4T1 cells described in **a**. **e** Luciferase assays of NF-κB activity in 4T1 cells described in **a** transfected with a NF-κB luciferase reporter or control vector. **f** Tumor weight 10 days after subcutaneous injection of 4T1 cells EMPTY or shMORG1 and shMORG2 ($n = 3$ BALB/c mice per group). **g** Percentage of immune system cells (gated on CD45$^+$ cells) detected in the primary tumor 10 days after injection of 4T1 EMPTY or shMORG1 and shMORG2 ($n = 3$ BALB/c mice per group). **h** Percentage of neutrophils detected in the primary tumor of each mouse 10 days after injection of 4T1. Bars in graphs represent standard errors (*$p < 0.05$; **$p < 0.01$; ***$p < 0.001$)

(Thermo Fisher Scientific). Human BT549 cells were cultured in RPMI medium (Thermo Fisher Scientific) with 0.023 IU ml$^{-1}$ insulin (Sigma Aldrich), 10% FCS (Euroclone) and Penicillin Streptomycin (Thermo Fisher Scientific). Human MCF10A cells were maintained in DMEM/F12 Ham's medium 1:1, 5% horse serum (Thermo Fisher Scientific), insulin (10 µg ml$^{-1}$, Sigma Aldrich), hydro-cortisone (0.5 µg ml$^{-1}$, Sigma Aldrich), epidermal growth factor (EGF 20 ng ml$^{-1}$, Sigma Aldrich) and Penicillin Streptomycin (Thermo Fisher Scientific). Human HeLa and murine E0771 cells were cultured in RPMI medium (Thermo Fisher Scientific) with 10% FBS (Euroclone) and 1% Penicillin Streptomycin (Thermo

Fisher Scientific). Murine NIH-3T3 cells were maintained in DMEM medium (Thermo Fisher Scientific) with 10% FCS (Thermo Fisher Scientific) and 1% Penicillin Streptomycin (Thermo Fisher Scientific).

Cell lines MCF10A, MCF7, BT549, MDA-MB-231, HEK293, NIH-3T3 and 4T1 were purchased from American Type Culture Collection and cultured under conditions specified by the provider. E0771 were obtained from Tebu-Bio. Mouse embryonic fibroblasts were prepared from mouse embryos[20]. Cells were routinely tested for mycoplasma contamination.

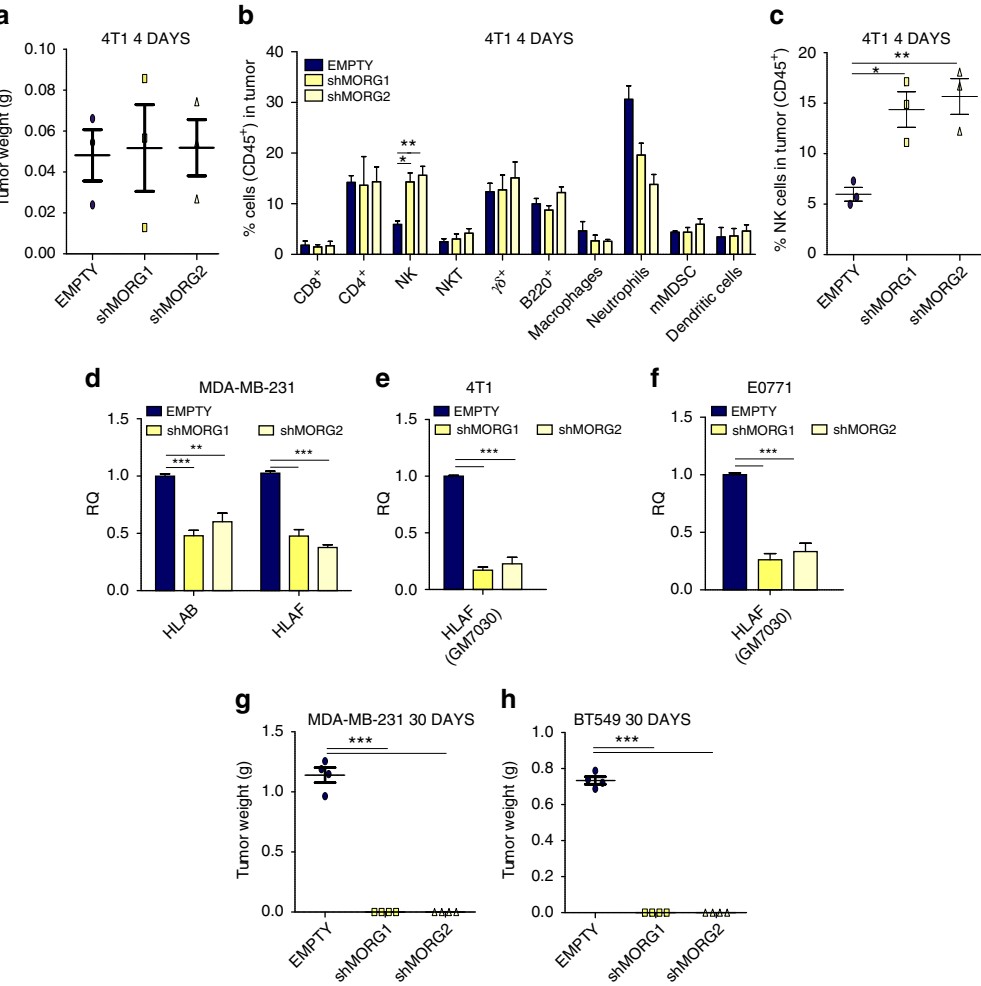

**Fig. 7** High Morgana levels in cancer cells induce NK cells recruitment at early phases of primary tumor formation. **a, b** Tumor weight **a** and percentage of immune cells (gated on CD45$^+$ cells) **b** 4 days after subcutaneous injection of 4T1 cells EMPTY or shMORG1 and shMORG2 ($n = 3$ BALB/c mice per group). **c** Percentage of natural killer (NK) cells detected in the primary tumor of each mouse 4 days after injection of 4T1. **d** Gene expression analysis by Real-time PCR of HLA-F and HLA-B in MDA-MB-231. **e, f** Gene expression analysis by Real-time PCR of GM7030 gene (mouse ortholog of HLA-F) in 4T1 **e** and E0771 **f** EMPTY or shMORG1 and shMORG2. **g** Graph showing the tumor weight 30 days after injection of MDA-MB-231 EMPTY or shMORG1 and shMORG2 ($n = 4$ nude mice per group). **h** Tumor weight 30 days after subcutaneous injection of EMPTY or shMORG1 and shMORG2 BT549 cells ($n = 4$ nude mice per group). Bars in graphs represent standard errors ($*p < 0.05$; $**p < 0.01$; $***p < 0.001$).

Morgana knockdown in MDA-MB-231, BT549, HeLa, HEK293, 4T1, and E0771 was performed by infecting cells with pGIPZ lentiviral particles expressing two different morgana shRNAs, together with the TurboGFP (Open Biosystems). BT549, MCF10A, MCF7, and NIH-3T3 overexpressing Morgana were obtained using a pLVX lentiviral vector coding for mouse Morgana fused to a myc tag. MMP9 overexpression was obtained infecting BT549 and MDA-MB-231 with PLX304-MMP9. HSP90 and CDC37 downregulation in MDA-MB-231 was performed by infecting cells with pLKO lentiviral particles expressing two different HSP90 or CDC37 shRNAs. IκBα silencing was obtained using pGIPZ lentiviral particles for MDA-MB-231 and pLKO viral particles for 4T1 and E0771.

**Plasmids and shRNA.** The following shRNAs were obtained from Open Biosystems: human CHORDC1 shRNAs V2LHS_24674 and V2LHS_24745; human IkBα shRNA V3LHS_410687, mouse CHORDC1 shRNAs V2THS_24746 and V2THS_24674. The following shRNAs were obtained from Sigma: human HSP90 shRNAs NM_007355.2-232s1c1 and NM_007355.2-2239s21c1; human CDC37 shRNAs NM_007065.3–819s1c1 and NM_007065.3-818s1c1, mouse IkBα shRNA NM_010907.2-235s21c1. IKK-2 WT (Addgene plasmid #11103) and IKK-2 S177E S181E (Addgene plasmid #11105) were gifts from Anjana Rao. p1242 3x-KB-L was a gift from Bill Sugden (Addgene plasmid #26699) and 3xAP1pGL3 (3xAP-1 in pGL3-basic) was a gift from Alexander Dent (Addgene plasmid #40342).

**Antibodies and reagents.** Western blotting and immunoprecipitations were performed using the following primary homemade antibody: a monoclonal antibody against Morgana (P1PP0) was generated in our laboratory using GST-

morgana fusion protein as an antigen[20]. Other commercially antibodies were used: Vinculin (Sigma, SAB4200080, 1:5000), CHORDC1 (Sigma, HPA041040, 1:1000), MMP9 (Abcam, ab76003, 1:1000), MMP2 (Santa Cruz, 8835, 1:500), HSP90 (Santa Cruz, 13119, 1:1000), P-IkBα Ser32-36 (Santa Cruz, 8404, 1:500), P-IkBα Tyr42 (Abcam, ab24783, 1:1000), IkBα (Cell Signaling, 4814, 1:1000), IKKα (Cell Signaling, 11930, 1:1000), IKKα (Cell Signaling, 2682, 1:100 for co-immunoprecipitation experiments), IKKβ (Cell Signaling, 8943, 1:1000), IKKγ (Santa Cruz, 8032, 1:500), αTubulin (Sigma, T5168, 1:8000), MBP (Cell Signaling, 2396, 1:1000), TNF-R1 (Santa Cruz, 8436, 1:500), TAK-1 (Santa Cruz, 166562, 1:500), p-AKT (Cell Signaling, 9271, 1:1000), AKT (Cell Signaling, 4691, 1:1000), CDC37 (Santa Cruz, 5617, 1:500), P-IKKα/β (Cell Signaling, 2697, 1:1000).

TNFα (300-01A and 315-01A) was purchased from Peprotech, the IKKβ inhibitor PS1145 (P6624), ROCK inhibitor Y27632 (Y0503), the PTEN inhibitor VO-OHpic (V8639), and the AKT inhibitor GSK69093 (SML0428) were from Sigma. BEZ235 (S1009) was purchased from Selleckchem.

**Lentiviral transduction.** Virus containing supernatants were collected 48 h after co-transfection of pCMV-VSV-G, pCMV Δ8.2 and the shRNA- or ORF-containing vector into HEK293T cells, and then added to the target cells. Cells were then selected with 10 μg ml$^{-1}$ puromycin[17].

**Anchorage-independent growth assay.** Soft agar colony formation assay on MDA-MB-231 was performed by resuspending cells in complete DMEM (Gibco-BRL) containing 0.3% low gelling agarose (Sigma-Aldrich) and seeding cells in triplicate into 6 well plates containing a 2 ml layer of solidified 0.6% agar (Sigma-

Aldrich). After 2 weeks colonies were stained with nitroblue tetrazolium (Sigma-Aldrich), photographed with Zeiss microscopy (Carl Zeiss) and colonies were counted using the ImageJ Software.

**Cell proliferation assay.** MDA-MB-231, BT549, 4T1, and E0771 cell growth curves were generated by plating $1 \times 10^4$ cells and staining cells with 0.1% crystal violet at the indicated times. After staining, wells were destained with 20% acetic acid and the absorbance of crystal violet solution was measured at 595 nm.

**Gelatin zymography.** For gelatin zymography, conditioned media and total protein extracts were collected from confluent MDA-MB-231 cells maintained in serum-free media for 24 h. Samples were subjected to electrophoresis using 10%

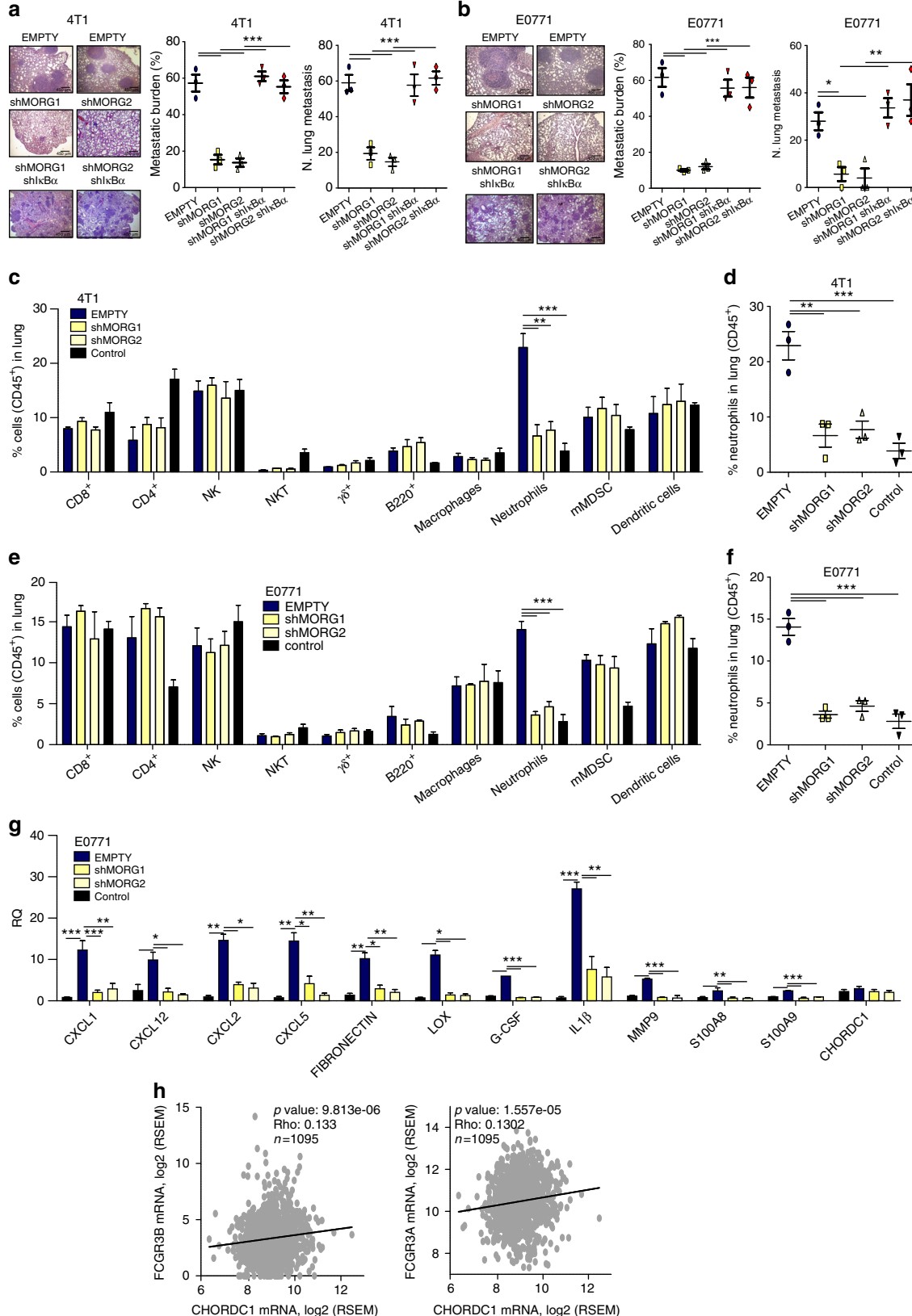

denaturing polyacrylamide gels containing 0.1% gelatin. Following electrophoresis, gel were renatured with 25% Triton in dH$_2$O and incubated at 37 °C for 16 h in a buffer composed of 500 mM Tris–HCl, pH 7, 8; 2 M NaCl and 50 mM CaCl$_2$. Then gels were stained with 0,5% Coomassie blue in 5% methanol, and 10% acetic acid in dH$_2$O and destained with 10% methanol, 5% acetic acid in dH$_2$O[58]. The bands on the gels were quantified using the ImageJ software.

**Immunoprecipitation and western blot analysis.** MDA-MB-231 or BT549 were lysed with lysis buffer (25 mM Tris–HCl pH 8, 1 mM MgCl$_2$, 10 mM NaF, 1 mM PMSF, 1 mM Na$_3$VO$_4$ and cocktail protease inhibitors (Roche)). After the lysis, cells were centrifuged at 1677 g for 5 min and then supernatants were ultra-centrifuged at 256365 g for 1 h. 500 ng of total protein extracts was incubated overnight at 4 °C with 5 µg of the selected antibody, then protein G-coated Sepharose was added for 45 min (GE Healthcare). Beads were washed 10 times in lysis buffer and resuspended in Laemmli buffer. For Western blot analysis, cells were lysed in Tris-buffered saline with 1% Triton X-100, plus phosphatase and protease inhibitors. Total protein extracts (30–50 µg) were analyzed by Western blotting[59] and detected by the chemiluminescent reagent LiteAblot (Euroclone). Band intensities were quantified using Quantity One software (Bio-Rad). Full scans of the most important Western blots are provided in Supplementary Fig. 12.

**Transwell invasion assay.** Invasion assays for MDA-MB-231 and BT549 were performed using BioCoat TM Matrigel Invasion Chambers with 8.0 µm pore size membrane (Becton Dickinson). Cells were previously starved for 24 h. Then, cells (75 × 10$^3$) were plated in the top chamber in medium without serum or growth factors, lower chambers were filled with complete growth media. After 24 h, the migrated cells present on the lower side of the membrane were fixed in 2.5% glutaraldehyde and stained with 0.1% crystal violet. For MDA-MB-231 transwell were photographed using an Olympus IX70 microscope. Invasion was evaluated by measuring the area occupied by migrated cells using the ImageJ software. For BT549 the invasion absorbance was measured after destaining of crystal violet with 20% acetic acid and the absorbance of crystal violet solution was measured at 595 nm.

**In vivo tumor and metastasis assays.** Experiments were carried out in accordance with the ethical guidelines of the European Communities Council Directive (2010/63/EU). Experimental approval was obtained from the Italian Health Ministry. Experimental metastasis assay and in vivo rescue experiment with IκBα shRNA were performed by injecting 5 × 10$^5$ MDA-MB-231 cells (in PBS) into the tail vein of 7-weeks-old female immunodeficient NSG mice (Charles River Laboratories). Mice were dissected 4 weeks later and macrometastases were counted at Nikon SMZ1000 stereomicroscope. Spontaneous metastases were evaluated in 7-weeks-old female immunodeficient NSG mice injected with 1 × 10$^6$ MDA-MB-231 cells (in PBS) and dissected 5 weeks later. Macrometastases were counted at Nikon SMZ1000 stereomicroscope. 1 × 10$^5$ 4T1 cells in 100 µl of PBS (mixed with Matrigel at 1:1 ratio) were injected subcutaneously in 6 weeks old female syngeneic BALB/c mice. After 4, 10, or 30 days mice were killed and the tumors were removed and weighed. 2 × 10$^5$ E0771 cells in 100 µl of PBS (mixed with Matrigel at 1:1 ratio) were injected subcutaneously in 6 weeks old female syngeneic C57BL/6 mice. After 6, 14 or 30 days mice were killed and the tumors were removed and weighed. For all metastasis studies, organs were formalin fixed and paraffin embedded, sectioned and haematoxylin and eosin (H&E) stained. Micrometastases were evaluated on specimens, with an Olympus BH2 microscope, on at least three different sections. Metastatic burden was evaluated as percentage of lung metastatic area using Metamorph Software (Universal Imaging Corporation), averaging at least six fields per sample. Experiments were performed in blind. 6 weeks old female immunocompromised nude mice were injected subcutaneously with 1 × 10$^6$ MDA-MB-231 or BT549 cells and after 30 days mice were killed and the tumors, if present, were removed and weighed.

**RNA isolation and Real-time PCR.** Total RNA from cells or tumors was isolated using Trizol Reagent (Thermo Fisher Scientific), following the manufacturer's recommendations. RNA was reverse transcribed by using Applied Biosystem high capacity cDNA reverse transcription kit. Gene expression analysis was performed using TaqMan Gene Expression Assays (Applied Biosystems) on an ABI Prism

7900HT sequence detection system (Applied Biosystems). We used 18S (Thermo Fisher) as the endogenous control throughout all experimental analysis. Analysis was performed using the $^{\Delta-\Delta}$Ct method to determine fold changes. We used gene-specific primers and the Universal Probe Library System (Roche Applied Sciences). Primers are listed in Supplementary Table 4. Cells were treated or not with TNF-α 10 nM 4 h before RNA extraction.

**Luciferase reporter assay.** Plasmids for the Luciferase Assay were purchased from Addgene: p1242-3x-KB-L, containing 3 NF-kappaB binding sites upstream of the Firefly Luciferase gene and 3xAP1pGL3, containing 3 AP-1 binding sites upstream of the Firefly Luciferase gene. For the Luciferase Assay 8 × 10$^4$ cells were plated on a 24-well plate. After 24 h, cells were co-transfected using Lipofectamine 2000 (Invitrogen) with 150 ng of pRL-TK Vector (PROMEGA) containing the *Renilla* luciferase construct, used as a normalizer and internal control, and with 650 ng of reporter vector (p1242-3x-KB-L or 3xAP1pGL3), or with empty vector pGL2 or pGL3, respectively (Promega). After 24 h trasfection cells were treated with TNF-α 10 nM and after 24 h Dual-Luciferase Reporter Assay were performed by Glomax instrument (Promega). Results are calculated as fold changes and shown as means of Firefly Luciferase activity normalized on Renilla luciferase activity.

**IKKβ kinase assay.** IKKβ activity was detected by performing a cold kinase assay[59]. MDA-MB-231 cells were lysed in IP buffer (Hepes 50 mM pH 7.6, NaCl 150 mM, EDTA 1 mM, 0.1% NP40, 10% glycerol, protease and phosphatase inhibitors), followed by centrifugation at 45,346×g for 15 min. IKKβ was immunoprecipitated from an equal amount of protein extracts and protein G beads were resuspended in kinase buffer (20 mM Hepes pH 7.6, 20 mM MgCl$_2$, 10 mM NaF, 0.1 mM Na$_3$VO$_4$, 5% glycerol, 1 mM DTT, Roche protease inhibitors cocktail and 1 mM PMSF). Then, we added ATP (10 µM) and 1 µg of IKBα substrate (Abcam) and 1 µg of MBP (Maltose Binding Protein) or 1 µg of MBP-Morgana or IKKβ inhibitor PS1145 20 µM to a volume of 25 µl at 30 °C for 10 min. Reactions were stopped by adding an equal volume of 2 × Laemmli buffer, followed by incubation at 95 °C for 10 min. Kinase activity was detected by Western blotting using P-IκBα and IκBα antibodies.

**Far western.** In total 1 µg of recombinant prey proteins was run on SDS–PAGE and transferred to nitrocellulose membrane. Proteins were denatured by incubating the membrane in the AC buffer (100 mM NaCl, 20 mM Tris pH 7, 6, 0.5 mM EDTA, 10% glycerol, 0.1% Tween 20, 2% skim milk powder and 1 mM DTT) containing 6 M guanidine-HCl for 30 min at RT. Then with the AC buffer containing 3 M guanidine-HCl for 30 min at RT. This is followed by incubation with the AC buffer containing 0.1 M and no guanidine-HCl at 4 °C for 30 min and 1 h, respectively[60]. After blocking with 5% milk for 1 h RT, the membrane was incubated with the bait recombinant protein MBP-Morgana overnight at 4 °C. Chemiluminescent detection was performed after the membrane incubation with the primary (1 h at RT) and the secondary antibody (1 h RT).

**Flow cytometry.** Single-cell suspensions from murine tumors and lungs were prepared by mechanical and enzymatic disruption in PBS with 1 mg ml$^{-1}$ collagenase P (Roche) for 1 h. Cells in suspension were filtered through a 40 µm cell strainer and centrifuged for 5 min at 120 g. After lysis of red blood cells, FcR were blocked with an anti-CD16/CD32 antibody (Becton Dickinson) and cells were stained with the indicated fluorochrome-conjugated antibodies (Supplementary Table 5) and Propidium Iodide (Sigma) to analyze viable cells. Lymphoid cell detection, samples were stained with anti-CD45, CD3, CD4, CD8, CD49b, B220, and γδ TCR antibodies, and % were calculated gating on the CD45 + population, while for myeloid and endothelial cell analysis samples were stained with anti-CD45, CD11b, F4/80, Ly6C, Ly6G, CD206, and CD31 and reported as percentage of myeloid cells on the CD45$^+$ population or of CD31 + endothelial cells on the CD45$^-$ population[61]. Dendritic cell (DC) subsets were stained with anti-CD45, CD11b, CD11c, CD24, CD64, IA/IE$^+$, and Ly6G antibodies[62]. The percentage of neutrophils was calculated as CD45$^+$, CD11b$^+$, F4/80$^-$, Ly6C$^+$, and Ly6G$^+$ cells. The percentage of natural killer cells (NK) was calculated as CD45$^+$, CD3$^-$ and CD49b$^+$, while percentage of NKT was calculated as CD45$^+$, CD3$^+$, and CD49b$^+$. For each analysis, a total of at least 20,000 CD45$^+$ living cells were analyzed. Flow

**Fig. 8** High Morgana levels in cancer cells induce neutrophil recruitment in the lung pre-metastatic niche. **a, b** Rapresentative haematoxylin and eosin stainings (left), quantification of lung metastatic area (middle) and number of lung metastases (right) of mice, 30 days after subcutaneous injection of 4T1 **a** and E0771 **b** EMPTY or shMORG1 and shMORG2 infected or not with IκBα shRNA (n = 3 BALB/c or C57BL/6 mice per group). **c** Percentage of immune cells detected in lungs of mice, 10 days after injection of 4T1 EMTPY or shMORG1 and shMORG2 cells (n = 3 BALB/c mice per group). **d** Percentage of neutrophils recruited in the lungs of each mouse 10 days after injection of 4T1. **e** Percentage of immune cells detected in lungs of mice, 14 days after injection of E0771 EMTPY or shMORG1 and shMORG2 cells (n = 3 C57BL/6 mice per group). **f** Percentage of neutrophils recruited in the lungs of each mouse 14 days after injection of E0771. **g** Gene expression analysis by Real-time PCR of neutrophil-attracting chemokines and pre-metastatic niche markers analyzed in lungs of mice 14 days after injection of E0771 EMPTY or shMORG1 and shMORG2 (n = 3 C57BL/6 mice per group). **h** Spearman correlation in mRNA levels between CHORDC1 (Morgana coding gene) and neutrophils marker (FCGR3A and FCGR3B). Bars in graphs represent standard errors (*p < 0.05; **p < 0.01; ***p < 0.001)

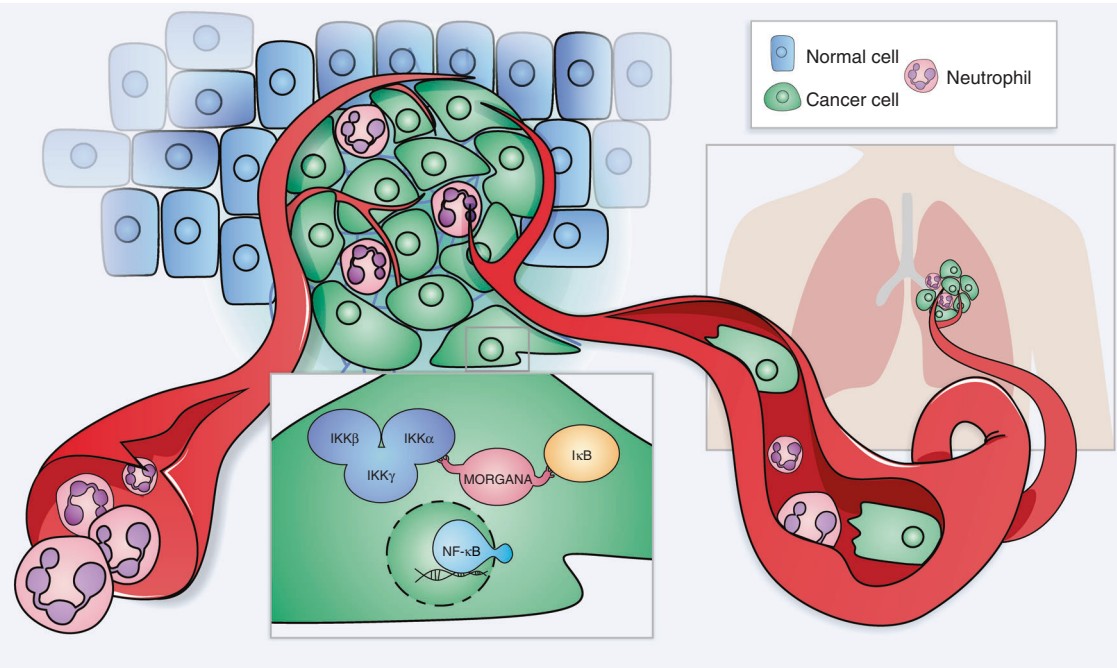

**Fig. 9** Schematic model based on our study showing an essential role for Morgana in connecting IKK complex and its substrate IκBα. The consequent activation of NF-κB in the tumor induces the production of several cytokines able to modulate the recruitment of cells of the immune system promoting metastatic spreading of tumor cells

cytometric analyzes were carried out on a BD FACSVerse using BD FACSSuite Software (Becton Dickinson).

**Gel filtration chromatography**. Gel filtration chromatography was performed on a Superose-6 10/300 GL column (GE Healthcare) using an AKTA purifier system (GE Healthcare). Protein extract was loaded onto the column and separated in gel filtration buffer (25 mM Tris–HCl pH 8.0, 1 mM MgCl$_2$, 10% glycerol), at a flow rate of 0.3 ml min$^{-1}$ and 0.5-ml fractions were collected. The molecular mass standards (GE Healthcare) used to calibrate the column were blue dextran (2000 kDa), thyroglobulin (669 kDa), ferritin (440 kDa), catalase (240 kDa) and aldolase (158 kDa). The fractions were directly analyzed by Western blotting after concentration using Nanosep 3 K or immunoprecipitated with anti-Morgana antibody and analyzed by Western blotting.

**Microarray**. Total RNA was extracted using Trizol and Absolute RNA miRNA kit (Agilent) following manufacturer's instructions. In total 1 μg of total RNA was amplified with ammino-allyl message amp II kit (Ambion) and labeled with Cy3 (GE-Heathcare), using the indirect labeling protocol. a total of 200 ng of labeled RNA were hybridized on Agilent SurePrint G3 Human Gene Expression 8 × 60 K glass arrays. The glass slide was scanned using an Agilent scanner and images were analyzed using Agilent Feature Extraction Software (version 10.7.3.1). Raw data were then processed using the Bioconductor package Limma (Linear models for microarray analysis). Differentially expressed transcripts were retrieved applying 0.01 as adjusted p-value cut-off and +1 or −1 as logFC cut-off for up or down-regulation, respectively.

**RNA preparation from tissue array specimens**. This study was conducted in compliance with the ethical regulatory requirements for the handling of biological specimens after appropriate informed consent and Institutional Review Board approval. The follow up study was performed on 152 samples of the tissue array previously described[14]. RNA from 58 human specimens was prepared using MasterPure complete DNA and RNA purification kit (Epicentre).

**Statistical significance**. For sample size calculation one tail Fisher test was used. Each experiment was repeated three times or more. The data are presented as means ± SEM. For statistical analyzes, significance was tested using a two-tailed Student's *t* test or, when required, one or two-way ANOVA with Bonferroni's correction. A minimum value of *p* < 0.05 was considered to be statistically significant. Statistical analyzes were performed using GraphPad Prism 4. The Mantel-Cox test was used in Kaplan-Meier survival curves.

**Statistics for spearman correlation analysis**. Spearman correlation analyzes were performed using the R language[63], Rstudio[64] suite and 'cor.test' function, method = 'spearman'. *P* values were adjusted for multiple testing with 'p.adjust'

function, method = 'fdr'. Gene expression data (HiseqV2) and protein RPPA-RBN data of TCGA BRCA, samples, version: 2015-02-24, were downloaded from UCSC Cancer Genome Browser[65–68] (http://genome-cancer.ucsc.edu). Gene expression data and metadata of 54 breast cancer cell lines[38] were downloaded from XENA Cancer Browser (xenabrowser.net).

**GSEA analysis**. Gene Set Enrichment Analysis[69, 70] was performed by running the GSEAPreranked tool from command line (gsea2-2.2.0.jar, with the following parameters: -mode Max_probe,-norm meandiv,-nperm 1000, -rnd_seed time-stamp,-set_max 500 and-set_min 15). The list's ranking metric was calculated on the Spearman's Rho correlation coefficient between CHORDC1 expression levels and all the genes present in the analyzed cohort (for what concern TCGA BRCA, 20530 genes in 1095 primary tumor samples cohort; while for Heiser's samples 18632 genes in 54 breast cancer cell lines). The gene sets used in the analysis, were downloaded (on May 16, 2016) from the Broad Institute GSEA website (http://software.broadinstitute.org/gsea/index.jsp), MSigDB database v5.2.

**Data availability**. Microarray data that support the findings of this study have been deposited in Gene Expression Omnibus with the accession code GSE86463. All other relevant data are available from the corresponding authors.

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

## Acknowledgements

We thank Prof. Roberto Piva for the kind gift of MMP9 expression vector. We would like to thank Flavio Cristofani and Antonellisa Sgarra for assistance with animal experiments. We thank Giuseppe Fragale for technical assistance. We thank Enzo Calautti for

comments on the manuscript. This work was supported by AIRC 2014 (IG 15880) to M. B., by AIRC (IG 16724) to F.C., by AIRC 2014 (IG 15217) to S.O., and by Progetto Giovani Ricercatori 2011 to A.M. F.F. was supported by a fellowship from FIRC (triennial fellowship "Cecilia Tocco").

## Author contributions

F.F. and M.B. designed the project and the experiments and wrote the paper. F.F. and L.S. performed the vast majority of the experiments. E.B., S.R., and C.R. performed experiments. A.K. and S.O. performed and analyzed the gel filtration experiments. E.M. and P. P. performed bioinformatics analysis. L.C. and F.C. performed and supervised, respectively, flow cytometry acquisition and analysis and contributed with discussions and advice. L.A. and I.C. supplied RNA from human cancer specimens and collected clinical follow-up data. M.M.-G., V.S., and G.C. performed microarray analysis. A.M. provided reagents and contributed with discussions and advice. L.S., F.A., E.T., S.O., G.T. provided discussions and advice. M.B. coordinated and directed the study.

## Additional information

**Competing interests:** The authors declare no competing financial interests.

