## [Peer Review File · Nature Communications]

Reviewers' comments:

Reviewer #1 (Remarks to the Author):

Fusella et al.

This paper reports that a protein (Morgana), previously described to be over-expressed in a portion of triple-negative breast cancers, drives NF- κ B activation to promote metastasis and neutrophil recruitment. The work is interesting as far as it goes, but more work is needed to relate the expression of Morgana more broadly in breast cancer (and other cancers) and to provide insight into the function of Morgana. Comments follow:

- 1) The authors published that Morgana is expressed at higher levels in a subset of TNBCs and also in other breast cancers. What about other cancers more broadly? In TNBC cells that don't express elevated levels of Morgana, does Morgana not drive NF- κ B activity in those cells? Does Morgana RNA expression correlate with the overexpression previously reported by this group?
- 2) Along the lines of pt. 1, in a HeLa cell (or potentially some other cell that has elevated NF- κ B activity) that doesn't express elevated levels of Morgana, does knockdown of Morgana block endogenous NF- κ B activation? What about TNF treatment of a non-transformed cell such as a fibroblast –is Morgana required for NF- κ B activation? In this regard, the authors provide references that Morgana is ubiquitously expressed (introduction).
- 3) The authors really don't acknowledge early on that Morgana has another name. In fact, they sometimes use the other name (CHORDC1).
- 4) Several groups have implicated NF- κ B as driving the tumor initiating cell phenotype in TNBC and Her2+ breast cancer. Does knockdown of Morgana block self-renewal of these cells (measured by sphere assays)?
- 5) Others have reported elevated PI3K and Akt regulatory proteins in TNBC. I didn't find compelling the evidence that PI3K and Akt are not involved downstream of Morgana to promote NF- κ B activity. The authors need to use a PI3K or Akt inhibitor to prove their point (related to effects of Morgana on ROCK I and II (refs. 14, 20).
- 6) If Morgana is important in driving the cell autonomous activation of NF- κ B, why do the authors use TNF in their studies?
- 7) It is interesting that Morgana appears to promote interaction of IKK with I κ B α . Given that others found that Morgana interacts with Hsp90 (typically important for IKK activity directed to I κ B α), does Hsp90 knockdown block recruitment of I κ B α and to IKK? What about cdc37? Does knockdown of Hsp90 block association of Morgana with IKK? Does knockdown of Hsp90 phenocopy the effect of Morgana knockdown (relative to NF- κ B activation, target gene expression, etc.). Thus, the association of Morgana with IKK could be through Hsp90, and Morgana could be the critical link in assembly of Hsp90 and cdc37 with the IKK complex. More work is needed here.

Reviewer #2 (Remarks to the Author):

In the manuscript "The IKK/NF- κ B signalling pathway requires Morgana to drive breast cancer metastasis", Fusella et al. identified Morgana as a novel component of the IKK complex required for NF- κ B signalling activation and breast cancer metastasis.

The findings concerning the role of Morgana in breast cancer are novel and interesting, and in this reviewer opinion, the work deserved publication in Nature Communication upon revision.

Particularly the characterization of Morgana as new component of NF- κ B signalling is novel and very well described in Fig. 4. In this reviewer opinion, to allow publications more experiments are needed to define NF- κ B signalling as the main function of Morgana related to metastasis. At the moment, some of the claims cannot be justified by the data. Moreover, it is not clear the impact of Morgana on the various tumorigenic properties, primary tumour growth, metastatic dissemination, metastasis colonization of the secondary site? The results show a bit of everything leaving the reader without a clear take home message. If the role of Morgana is to sustain NF- κ B signalling in these various tumorigenic activities, it needs to be better defined. Finally, while the experiments are overall well performed, the rationale for some experiments in the text is not always clear or in line with the current knowledge.

Comments:

1. The data in Fig 1c and Supp Fig 1c, using shRNAs, provide the evidence that Morgana is important for luminal breast cancer cells invasion. In *J Pathol* 234, 152-163 (2014), the same authors have shown how Morgana boosts AKT activity and survival in breast cancer cells by downregulating PTEN via ROCK I inhibition. In this 2014 study, the authors confirmed Morgana inhibitory activity on ROCK I by a decrease in pMLC levels in the cells. Considering the known requirement of ROCK I and pMLC for cancer cell motility (*Cancer Cell*. 2003 Jul;4(1):67-79.), its inhibition is generally associated to decrease of cancer cell invasion also in breast cancer cells (*Oncotarget*. 2016 Nov 8;7(45):73593-73606, *Clin Exp Med*. 2016 Feb;16(1):37-47.). Indeed therapeutic interventions with ROCK kinase inhibitor are considered to decrease cancer cell migration and invasion (*Clin Cancer Res*. 2015 Dec 1; 21(23): 5209–5214). Therefore, in line with the known positive function of ROCK for cancer cell migration, the previously reported inhibitory activity of Morgana on ROCK is in contrast with the results shown in Figure 1c and Supp Figure 1c. The authors, do not comment on this controversy between the previous findings and this effect on invasion, instead use the unlikely possibility that the reduced migration observed in shMorgana cells might depend on ROCK inhibition as the rationale for the experiments in Supp Fig 1d,e, where they formally test the unexpected hypothesis that ROCK inhibition could be causing a boost in invasion. Even if the results of the test show no link between these two activities, the data should be commented and presented differently. The text should describe the data based on the large body of literature of ROCK activity in cancer invasion.

Moreover, Supp Fig 1d,e shows that Y27632 inhibitor fails to block breast cancer cells invasion. Importantly, opposite results were shown in other studies using the same inhibitor (*Cell Commun Signal*. 2014 Oct 5;12:54.; *Anticancer Agents Med Chem*.; Ghasemi a. et al., *Cell Signal*. 2017 Jan 16; *Cell Mol Biol* 2016 Oct 31;62(12):91-96.; *Oncogene*. 2014 Dec 4;33(49):5582-91; *Cancer Res*. 2009 Nov 15;69(22):8742-51.; etc) therefore the authors need to comment on these inconsistencies.

2. The final message of the paper is that Morgana, by sustaining NF- κ B activation, is required for breast cancer metastasis. However remain unclear what is the specific step of the metastatic process that is affected and some of the mechanistic links between the various results are missing. Rescuing experiments are required to draw the conclusions of the manuscript.

i. The reduction in metastatic activity of shMorgana tumour is shown in figure 1e-g. This is a key experiment, but the link with NF κ B signalling needs to be validated with a rescuing experiment similar to Fig 3k-m.

ii. Initially Fig 1 shows that Morgana is required for cancer cell invasion and Fig 2 suggest that MMP9 expression would be the functional mediator: can exogenous MMP9 expression (or NF κ B stabilization) rescue the invasion deficit of shMorgana cells?

iii. Invasion is the first step of metastasis, but Fig 1h,i show that when the invasion step is overcome by seeding the cancer cells directly at metastatic site, Morgana reduced the ability of cancer cells to grow specifically there. This metastatic colonization and grow effect is robustly proven to be linked to Morgana activity on NF- κ B in the rescue experiment of Figure 3k-m. This effect alone would justify the reduction of metastasis independently of the effect on cancer cell

invasion, but it need to be better link to the primary tumour growth.

iv. When testing Morgana effect in mouse syngeneic model, primary tumour formation, similarly to secondary growth of human cell line in NSG mice, is inhibited. Firstly, to clarify the link with NF- κ B, again the authors need to show that this effect on primary tumour growth and neutrophils infiltration is rescued by NF- κ B stabilization. Given the previous report about the neutrophil recruitment and IL1beta secretion in breast cancer (Nature. 2015 Jun 18;522(7556):345-8.) it would be easy to discuss the role of NF- κ B in neutrophils recruitment. Secondly, the data presented in Figure 7 show a reduction in primary tumour growth in both 4T1 and E0771. With those differences in primary tumours no conclusion can be made of the number of spontaneous metastasis. Metastasis assessment needs to consider the difference in primary tumour size and progression. Similarly, the comparison of the tumour microenvironment in tumours at completely different stages cannot be made: the difference in immune cell infiltration could only be reflecting the difference in size. Authors should look at the tumour microenvironment at early stages of the tumour growth when the tumour sizes are still comparable.

3. Experiment in Fig 1h-I assesses spontaneous metastasis from primary tumours, but state that primary tumour growth is not affected by referencing a previous study and an independent experiment. The paper needs to show to the primary tumour growth of the very same experiment where metastasis is assessed.

4. Fig3 introduce the functional link between MMP9 expression, required for invasion activity, and Morgana. Figure 4 links the NF- κ B signalling to MMP9 expression, which form the rational to investigate Morgana activity on NF- κ B signalling. In Fig 2g,h, overexpression of Morgana in cells that do not express it show increase in MMP9, only after TNFalpha stimulation. The rational of this is not clear at this stage as the impact of Morgana on NF- κ B activity only discussed with the data in Fig 3. To help the flow of the data, Fig 2g,h should be moved in Figure 3, this would help the construction of the link with NF- κ B signalling. Also the effect of Morgana overexpression on MMP9 levels without stimulation should be shown and discussed.

5. Fig3: one of the two cell lines of each experimental settings can go in supplement, the information are redundant.

Fig 6 only confirmed the consistency of the effect of Morgana depletion in two mouse breast lines; the entire figure should be a supplementary figure.

6. It is puzzling the different effect of Morgana depletion on the primary tumour growth in human versus mouse breast cancer cells. This difference is unlikely simply due to an effect of the adaptive immune system as Fig 7c,d show no difference in CD4 or CD3 cells. However it could be due to the neutrophils infiltration of the primary tumour. Indeed this hypothesis cannot be tested in NSG mice that also have defects in the adaptive immunity. It would be desirable to test if the human cell lines show a difference in primary tumour growth in RAGko or Nude mice, where the innate immunity is maintained functional. Here a difference in neutrophils would be observed with the effect on tumour growth, and the functional relevance of the neutrophils infiltration for primary tumour growth could be estimated.

Reviewer #3 (Remarks to the Author):

In the manuscript by F. Fusella et.al., the authors demonstrate an essential role for Morgana in promoting breast cancer metastasis. They show that high Morgana correlates with NF- κ B target gene expression and augment cytokines promoting neutrophil recruitment which favors the promotion of primary tumor growth and pre-metastatic niche formation in the lung. The work is very novel and its relevance is extremely timely for advancing knowledge in breast

cancer metastasis to the lung.

There are comments to be made.

Importantly, the authors need to improve the connections of their studies throughout the manuscript. For instance, the authors introduce molecules, such as IL-1B, IL-6, CCL5 which was analyzed extensively with 4T1 and E0771 in Figure 6 but not analyzed in a similar fashion for MDA-231 studies. Proliferation studies for BT549 in supplementary figure 1 are not performed for MDA-231 in Figure 1 C/d. HSP90 introduced in the earlier figures are not carried throughout the entire result section.

Also, the focus on MMP-9 was a bit abrupt in its introduction in figure 2. To validate MMP-9 as an essential molecule related to Morgana, the authors should knockout MMP9 in tumor cells to determine if Morgana knockdown or overexpression could influence metastasis in the absence of MMP9 manipulation.

The results could be more convincing if the authors overexpress Morgana in the BT549 cell line which is known not to metastasize.

There are other comments by figures.

Figure 1:

The authors describe MDA-231 metastasis to the liver, heart and kidney, these sites are not well-described sites of metastasis.

The authors need to comment since this is a new description of this tumor spreading to these unusual sites.

Figure 2:

It is unclear if Morgana is always internal or does the protein get externalized/secreted? In C, D, include MMP-2 and Morgana for both.

quantify metastatic burden in e.

In G, h, the authors used TNF-a in the assays, the same should be the same for e and f studies.

Figure 3:

show SP-1 in c, d

Figure 4:

repeat western blots for clarity in H and I.

Figure 5:

the survival data needs to be consistent for a and b with the same Y axis blots.

Figure 6:

include ShMORG2 for a and for e.

Need to be consistent

for g and h, need CCL5, IL-6 for both tumor cell types

where is TNFa here

Figure 7:

for cell characteristics, the authors should consider B cells, NK cells, dendritic cells, endothelial cells in these studies.

The pre-metastatic niche work is very fascinating and should be further explored. How do the neutrophils get recruited to the lungs. There are previous publications on MMP-9 in the pre-metastatic niche, does the authors observe a similar increase in MMP-9? What about IL-6 in the pre-metastatic niche. What happens to ECM changes which can be affected by MMP-9

Point-by-point answers to the reviewers (The reviewers' comments are in Italics. Answers are non-italicized).

We thank the reviewers for their considerations and for their helpful suggestions.

In order to make easier the revision the changes in the text are in red.

Reviewer #1:

1) The authors published that Morgana is expressed at higher levels in a subset of TNBCs and also in other breast cancers. What about other cancers more broadly? In TNBC cells that don't express elevated levels of Morgana, does Morgana not drive NF- κ B activity in those cells? Does Morgana RNA expression correlate with the overexpression previously reported by this group?

To evaluate the relevance of Morgana overexpression in other cancers, we performed a Morgana expression analysis on the 22 TCGA tumours for which data on adjacent normal tissue were present. Morgana is significantly upregulated in 40% (9 out of 22) and downregulated in 4% (1 out of 22) tumour types, compared to normal tissue. The results are shown in Supplementary Fig. 7g, h (page 12).

In order to study the correlation between Morgana expression and NF- κ B gene signature in breast cancer cell lines, we performed a gene set enrichment analysis on a panel of breast cancer cell lines¹, as we previously did for the TCGA BRCA cohort (Fig. 5d). The results showed that in the whole panel of breast cancer cell lines as well as in the triple negative cancer (TNBC) cell line subset (coloured circles in Supplementary Fig. 7e), Morgana expression levels positively correlated with the NF- κ B gene signatures (Supplementary Fig. 7f). This implies that in TNBC cell lines expressing low Morgana levels, NF- κ B gene signature is significantly less expressed compared to high Morgana expressing cells (page 12).

We performed a Real-time PCR on 58 human breast cancer specimens that we previously analyzed by immunohistochemistry (IHC)². We found a positive correlation between Morgana protein expression and its mRNA levels (Supplementary Fig. 7a, page 11).

We used the same RNA samples to enrich (from n=22 to n=58) the correlation analysis on MMP9 and Morgana expression levels showed in Fig. 5h (page 12).

2) Along the lines of pt. 1, in a HeLa cell (or potentially some other cell that has elevated NF- κ B activity) that doesn't express elevated levels of Morgana, does knockdown of Morgana block endogenous NF- κ B activation? What about TNF treatment of a non-transformed cell such as a

fibroblast –is Morgana required for NF-κB activation? In this regard, the authors provide references that Morgana is ubiquitously expressed (introduction).

We downregulated Morgana in HeLa cells, as suggested by the Reviewer, and we observed a decrease both in IκBα phosphorylation (Supplementary Fig. 5c, page 9) and in the expression of NF-κB target genes evaluated by Real-time PCR (Supplementary Fig. 4c, page 8). Also in mouse embryonic fibroblasts (MEFs) obtained from *morgana* +/- mice³, we observed a reduction in IκBα phosphorylation (Supplementary Fig. 5f, page 9), NF-κB transcriptional activity (Supplementary Fig. 3i, page 8) and NF-κB target gene expression (Supplementary Fig. 4f, page 8) if compared to wild type MEFs. The differences between *morgana* +/- and wild type MEFs remained significant in presence or absence of TNF-α treatment (Supplementary Fig. 4f, page 8) (Supplementary Fig. 5f, page 9).

As mentioned by the Reviewer, Morgana is ubiquitously expressed. Accordingly, our new data further sustain that NF-κB signalling depends on Morgana expression levels not only in breast cancer cell lines, but also in other cell types, as in a non-breast cancer cell line (HeLa), in immortalized human and mouse cells (HEK293 and NIH-3T3) and in primary mouse embryonic fibroblasts (MEFs) (Supplementary Fig. 3g-i, Supplementary Fig. 4c-f, Supplementary 5c-f and Supplementary Fig.6a,b).

3) The authors really don't acknowledge early on that Morgana has another name. In fact, they sometimes use the other name (CHORDC1).

We apologize for the missing information. Morgana is the protein name, while CHORDC1 is the Morgana coding gene. We added this information in the abstract and in page 4.

4) Several groups have implicated NF-κB as driving the tumor initiating cell phenotype in TNBC and Her2+ breast cancer. Does knockdown of Morgana block self-renewal of these cells (measured by sphere assays)?

To perform this experiment we choose three breast cancer cell lines: MCF7 (ER+, PgR+), T47D (Her2+), HCC1937 (triple negative), based on their ability to propagate as long-term mammosphere cultures⁴. Despite the reports involving NF-κB canonical pathway with the tumour initiating cell phenotype^{5,6,7,8}, Morgana downregulation did not alter tumoursphere formation ability (Fig. a below, t1: tumourspheres at the first passage and t2: tumourspheres at the second passage). Note that Morgana downregulation in the three different breast cancer cells inhibited IκBα phosphorylation (Fig. b-d below). These results are of interest, since they suggest that Morgana can

impact on specific NF- κ B target genes, thus regulating particular features of cancer cells. Alternatively, additional signalling may be concomitantly required to NF- κ B activation to drive tumour initiating cell phenotype in breast cancer cells^{9,10}.

We believe that these results need a dedicated study to dissect the precise molecular details through which Morgana can impact on cancer cell stemness. However, if the Reviewer believes that the results are anyway relevant for this work, we can add the following sentences in the Discussion section:

“Despite the reports involving NF- κ B canonical pathway with the tumour initiating cell phenotype^{5,6,7,8}, Morgana downregulation did not alter tumoursphere formation ability measured by sphere assays (data not shown) on three different breast cancer cell lines (MCF-7, T47D, HCC1937)⁴. Further work is needed to understand if Morgana regulation on NF- κ B pathway impacts on specific target genes or if the concomitant activation of parallel pathways is needed to sustain cancer stem cell phenotype^{9,10}”

5) Others have reported elevated PI3K and Akt regulatory proteins in TNBC. I didn't find compelling the evidence that PI3K and Akt are not involved downstream of Morgana to promote NF- κ B activity. The authors need to use a PI3K or Akt inhibitor to prove their point (related to effects of Morgana on ROCK I and II (refs. 14, 20)).

We treated MCF10A and MCF7 overexpressing Morgana and control cells with the PI3K inhibitor BEZ235 and the AKT inhibitor GSK690693 for 24h and no significantly changes in NF- κ B transcriptional activity were detected by luciferase assays (Supplementary Fig. 3c-f, page 8).

6) *If Morgana is important in driving the cell autonomous activation of NF- κ B, why do the authors use TNF in their studies?*

We now performed all the analysis on NF- κ B target genes with and without TNF- α and we obtained significant results in both conditions (Fig. 3c,d, Supplementary Fig. 4a-f and Supplementary Fig. 8g,h).

7) *It is interesting that Morgana appears to promote interaction of IKK with I κ B α . Given that others found that Morgana interacts with Hsp90 (typically important for IKK activity directed to I κ B α), does Hsp90 knockdown block recruitment of I κ B α and to IKK? What about cdc37? Does knockdown of Hsp90 block association of morgana with IKK? Does knockdown of Hsp90 phenocopy the effect of Morgana knockdown (relative to NF- κ B activation, target gene expression, etc.). Thus, the association of Morgana with IKK could be through Hsp90, and Morgana could be the critical link in assembly of Hsp90 and cdc37 with the IKK complex. More work is needed here.*

As the Reviewer pointed out, it has been demonstrated that Hsp90 and cdc37 take part to the IKK complex. We downregulated the constitutive Hsp90 β isoform and the co-chaperone cdc37 using in both cases two different shRNAs and we were not able to detect a decrease in the association between Morgana and IKK or I κ B α (Supplementary Fig. 6f,g, page 10). It is to note that Hsp90 is essential for cell survival and cancer cells are often addicted to Hsp90¹¹. Accordingly, MDA-MB-231 cells hardly survive after Hsp90 and cdc37 silencing. Moreover, Hsp90 inhibition cause degradation of numerous proteins involved in signal transduction. For all these reasons the phenotype of Hsp90 silenced cells is definitely more dramatic if compared to Morgana interfered cells and in our opinion would be difficult to discriminate among NF- κ B related phenotypes and side effects induced by massive protein degradation. Moreover, our results demonstrated that Morgana is not required for Hsp90 recruitment to the IKK complex, since in cells downregulated for Morgana, Hsp90 is still present in the high molecular weight IKK complex in gel filtration experiments (Fig.4i, right panel). In addition, we detected a direct association between Morgana and I κ B α by Far Western analysis (Supplementary Fig.6c, page 9), indicating that Hsp90 is not involved in this binding.

Reviewer #2

1. *The data in Fig 1c and Supp Fig 1c, using shRNAs, provide the evidence that Morgana is important for luminal breast cancer cells invasion. In J Pathol 234, 152-163 (2014), the same authors have shown how Morgana boosts AKT activity and survival in breast cancer cells by downregulating PTEN via ROCK I inhibition. In this 2014 study, the authors confirmed Morgana*

inhibitory activity on ROCK I by a decrease in pMLC levels in the cells. Considering the known requirement of ROCK I and pMLC for cancer cell motility (Cancer Cell. 2003 Jul;4(1):67-79.), its inhibition is generally associated to decrease of cancer cell invasion also in breast cancer cells (Oncotarget. 2016 Nov 8;7(45):73593-73606, Clin Exp Med. 2016 Feb;16(1):37-47.). Indeed therapeutic interventions with ROCK kinase inhibitor are considered to decrease cancer cell migration and invasion (Clin Cancer Res. 2015 Dec 1; 21(23): 5209–5214). Therefore, in line with the known positive function of ROCK for cancer cell migration, the previously reported inhibitory activity of Morgana on ROCK is in contrast with the results shown in Figure 1c and Supp Figure 1c.

The authors, do not comment on this controversy between the previous findings and this effect on invasion, instead use the unlikely possibility that the reduced migration observed in shMorgana cells might depend on ROCK inhibition as the rationale for the experiments in Supp Fig 1d,e, where they formally test the unexpected hypothesis that ROCK inhibition could be causing a boost in invasion. Even if the results of the test show no link between these two activities, the data should be commented and presented differently. The text should describe the data based on the large body of literature of ROCK activity in cancer invasion. Moreover, Supp Fig 1d,e shows that Y27632 inhibitor fails to block breast cancer cells invasion. Importantly, opposite results were shown in other studies using the same inhibitor (Cell Commun Signal. 2014 Oct 5;12:54.; Anticancer Agents Med Chem.; Ghasemi a. et al., Cell Signal. 2017 Jan 16; Cell Mol Biol 2016 Oct 31;62(12):91-96.; Oncogene. 2014 Dec 4;33(49):5582-91; Cancer Res. 2009 Nov 15;69(22):8742-51.; etc) therefore the authors need to comment on these inconsistencies.

We agree with the Reviewer, we commented based on the results in the literature and presented the data in a different way (page 6). Regarding to old Supplementary Fig. 1d, we apologize for inaccuracy. We focalized our attention to the significant differences among empty and shMorgana cells, however a significant reduction between empty untreated and treated with 2 microM Y27632 is present, in line with the papers cited by the Reviewer. In the new Supplementary Fig.2b we added this information.

2. The final message of the paper is that Morgana, by sustaining NF- κ B activation, is required for breast cancer metastasis. However remain unclear what is the specific step of the metastatic process that is affected and some of the mechanistic links between the various results are missing. Rescuing experiments are required to draw the conclusions of the manuscript.

i. The reduction in metastatic activity of shMorgana tumour is shown in figure 1e-g. This is a key experiment, but the link with NFκB signalling needs to be validated with a rescuing experiment similar to Fig 3k-m.

As requested by the Reviewer, we demonstrated that the ability of primary tumour to form metastasis depends on NF-κB activity. In particular, we performed a rescuing experiment by downregulating IκBα in MDA-MB-231 shMorgana cells and injecting them subcutaneously in NSG mice. The restoration of NF-κB activity completely rescues the impairment in metastasis formation in Morgana downregulated cells (Fig. 3j-l, page 9), as already demonstrated in experimental metastasis assay (Fig. 3h,i).

ii. Initially Fig 1 shows that Morgana is required for cancer cell invasion and Fig 2 suggest that MMP9 expression would be the functional mediator: can exogenous MMP9 expression (or NFκB stabilization) rescue the invasion deficit of shMorgana cells?

We performed the experiment requested by the Reviewer and our new data indicate that MMP9 overexpression is able to rescue the impairment in cell invasion in both MDA-MB-231 and BT549 downregulated for Morgana (Fig. 2i,j, page 7).

iii. Invasion is the first step of metastasis, but Fig 1h,i show that when the invasion step is overcome by seeding the cancer cells directly at metastatic site, Morgana reduced the ability of cancer cells to grow specifically there. This metastatic colonization and grow effect is robustly proven to be linked to Morgana activity on NF-κB in the rescue experiment of Figure 3k-m. This effect alone would justify the reduction of metastasis independently of the effect on cancer cell invasion, but it need to be better link to the primary tumour growth.

As pointed out by the Reviewer, cancer metastasis is a multistage event, comprising local invasion, vessel intravasation, survival in suspension, extravasation, survival in foreign organs and re-entering in the cell cycle, in which NF-κB signalling plays multiple roles¹². We performed rescue experiments with shIκBα in both experimental and spontaneous metastasis assays (Fig. 3h-l, page 9), as well as in primary tumour growth and neutrophil recruitment in syngeneic mouse models (Fig. 6a-e and Supplementary Fig. 9a,d-g, page 13). In all cases, we demonstrated a causal involvement of NF-κB. As also suggested by the Reviewer, we believe that Morgana sustains NF-κB signalling in the different steps of metastasis formation.

iv. When testing Morgana effect in mouse syngeneic model, primary tumour formation, similarly to secondary growth of human cell line in NSG mice, is inhibited. Firstly, to clarify the link with NF- κ B, again the authors need to show that this effect on primary tumour growth and neutrophils infiltration is rescued by NF- κ B stabilization. Given the previous report about the neutrophil recruitment and IL1beta secretion in breast cancer (Nature. 2015 Jun 18;522(7556):345-8.) it would be easy to discuss the role of NF- κ B in neutrophils recruitment.

As suggested by the Reviewer, we performed I κ B α downregulation in 4T1 and E0771 silenced for Morgana and we obtained a total rescue in primary tumour growth, neutrophil recruitment (Fig. 6a-e and Supplementary Fig. 9a,d-g) and metastasis formation (Fig. 8a,b). We discussed the role of NF- κ B signalling in the recruitment of neutrophils in primary tumour in the discussion section (page 17).

Secondly, the data presented in Figure 7 show a reduction in primary tumour growth in both 4T1 and E0771. With those differences in primary tumours no conclusion can be made of the number of spontaneous metastasis. Metastasis assessment needs to consider the difference in primary tumour size and progression. Similarly, the comparison of the tumour microenvironment in tumours at completely different stages cannot be made: the difference in immune cell infiltration could only be reflecting the difference in size. Authors should look at the tumour microenvironment at early stages of the tumour growth when the tumour sizes are still comparable.

We analyzed tumour microenvironment at three different time points from cancer cell injections for both 4T1 and E0771 cells. We choose slightly different timings for the two cell lines, given their subtle differences in primary tumour growth and metastasis formation. In the first and second time points (4 and 10 days for 4T1 and 6 and 14 days for E0771), the sizes of Morgana downregulated and control tumours were comparable allowing the analysis requested by the Reviewer. Neutrophil recruitment was robustly reduced by Morgana silencing at the second time points (10 and 14 days from injection of 4T1 and E0771, respectively), while no differences were detected earlier (Fig.6 f-h and Supplementary Fig. 10a-c, page13). In addition, at the first time point (4 and 6 days from injection of 4T1 and E0771, respectively), we noticed a significant increase in natural killer (NK) cells in Morgana downregulated tumours (Fig. 7a-c and Supplementary Fig. 10d-f, page 13). No differences were found in other immune cells (Fig. 6b,g, Fig.7b, Supplementary Fig. 9d, Supplementary Fig. 10b,e and Supplementary Fig. 11a,b, page 13) and endothelial cells at all time points (Supplementary Fig. 11c,d, page 13). We also repeated the analysis at 30 days from injection for both cell lines in which primary tumour sizes were significantly different depending on Morgana expression levels (Fig. 6a-c and Supplementary Fig.9a,d,e, page 12). We are grateful to the reviewer

because thanks to her/his suggestion we better characterized the cell types present in the tumour microenvironment at different time points.

3. Experiment in Fig 1h-I assesses spontaneous metastasis from primary tumours, but state that primary tumour growth is not affected by referencing a previous study and an independent experiment. The paper needs to show to the primary tumour growth of the very same experiment where metastasis is assessed.

We added the data requested in Fig. 1j.

4. Fig3 introduces the functional link between MMP9 expression, required for invasion activity, and Morgana. Figure 4 links the NF- κ B signalling to MMP9 expression, which form the rational to investigate Morgana activity on NF- κ B signalling. In Fig 2g,h, overexpression of Morgana in cells that do not express it show increase in MMP9, only after TNFalpha stimulation. The rational of this is not clear at this stage as the impact of Morgana on NF- κ B activity only discussed with the data in Fig 3. To help the flow of the data, Fig 2g,h should be moved in Figure 3, this would help the construction of the link with NF- κ B signalling. Also the effect of Morgana overexpression on MMP9 levels without stimulation should be shown and discussed.

We agree with the Reviewer, in Fig. 2g,h we presented data on MMP9 and MMP2 expression without TNF- α treatment, while the expression analysis with TNF- α (old Fig. 2g,h) has been moved in Fig. 3d and Supplementary Fig. 4b. In both conditions, we obtained significant differences between cells overexpressing Morgana and controls.

5. Fig3: one of the two cell lines of each experimental settings can go in supplement, the information are redundant.

We moved the experiments on BT549 and MCF10A to Supplementary Fig. 3a,b.

Fig 6 only confirmed the consistency of the effect of Morgana depletion in two mouse breast lines; the entire figure should be a supplementary figure.

As requested by the Reviewer, we moved all the experiments previously shown in Fig. 6 in Supplementary Fig. 8.

6. It is puzzling the different effect of Morgana depletion on the primary tumour growth in human versus mouse breast cancer cells. This difference is unlikely simply due to an effect of the adaptive immune system as Fig 7c,d show no difference in CD4 or CD3 cells. However it could be due to the

neutrophils infiltration of the primary tumour. Indeed this hypothesis cannot be tested in NSG mice that also have defects in the adaptive immunity. It would be desirable to test if the human cell lines show a difference in primary tumour growth in RAGko or Nude mice, where the innate immunity is maintained functional. Here a difference in neutrophils would be observed with the effect on tumour growth, and the functional relevance of the neutrophils infiltration for primary tumour growth could be estimated.

We thank the Reviewer for the suggestion. We injected MDA-MB-231 and BT549 cells in nude mice and we obtained even stronger differences in primary tumour growth than in syngeneic mice between Morgana downregulated cells and controls. In fact, while control cells grow subcutaneously in nude mice (n=3/3), cells downregulated for Morgana do not form tumour at all (n=6/6) (Fig. 7g,h, page 13).

In this revised version of the manuscript we also investigated the role of NK cells in the tumour microenvironment and we found a significant increase of this population at early stages of tumour development (6 days for E0771 and 4 days for 4T1) (Fig.7a-c and Supplementary Fig.10d-f).

The ability of NK cells to recognize and kill cancer cells depends on the balance between activating or inhibitory signals. MHC class I receptors on cancer cells bind to NK receptors and potentially inhibit their activation¹³. The high recruitment of NK cells in tumours in which Morgana has been silenced may be due to the downregulation of MHC class I expression, as highlighted by microarray and Real-time PCR analysis on MDA-MB-231, 4T1 and E0771 interfered for Morgana vs control cells (Fig.7d-f, page 13). The total impairment of MDA-MB-231 shMorgana cells to grow in nude mice did not allow a neutrophil recruitment analysis. However, given that NK cells are present and highly functional in nude mice, we hypothesize that these cells are responsible for the lack of engraftment of shMorgana tumours in this system and, at least partially, for the defective growth of shMorgana tumours in syngeneic mice (discussed in page 16-17).

Reviewer #3

Importantly, the authors need to improve the connections of their studies throughout the manuscript. For instance, the authors introduce molecules, such as IL-1B, IL-6, CCL5 which was analyzed extensively with 4T1 and E0771 in Figure 6 but not analyzed in a similar fashion for MDA-231 studies. Proliferation studies for BT549 in supplementary figure 1 are not performed for MDA-231 in Figure 1 C/d. HSP90 introduced in the earlier figures are not carried throughout the entire result section.

We apologize for inconsistencies throughout the manuscript. In the revised version we standardized the NF- κ B target genes analyzed by Real-time PCR (MMP9, CCL5, IL-1 α , IL-1 β , IL-6 and TNF-

α) in the different cell lines (Fig. 3c,d, Supplementary Fig. 4 and Supplementary Fig. 8g,h). We added proliferation studies for MDA-MB-231 in Fig. 1c. We showed Hsp90 in all experiments in which we analyzed the IKK complex composition (Fig. 4g,h,i, Supplementary Fig. 5a,b). Moreover, we investigated the role of Hsp90 in Morgana binding to I κ B α and IKK complex (Supplementary Fig. 6f).

Also, the focus on MMP-9 was a bit abrupt in its introduction in figure 2. To validate MMP-9 as an essential molecule related to Morgana, the authors should knockout MMP9 in tumor cells to determine if Morgana knockdown or overexpression could influence metastasis in the absence of MMP9 manipulation.

We improved the introduction in the second paragraph of the Result section (page 7). To validate MMP-9 as a crucial molecule in Morgana-dependent invasion in breast cancer cells, we knockdown MMP-9 in MDA-MB-231 and we overexpressed it in MDA-MB-231 and BT549 silenced for Morgana. Invasion assays were performed and the results are shown in Fig. 2i,j. MMP-9 downregulation reduced invasion abilities of MDA-MB-231 and its overexpression, in both MDA-MB-231 and BT549 silenced for Morgana, rescued invasion impairment (page 7).

The results could be more convincing if the authors overexpress Morgana in the BT549 cell line which is known not to metastasize.

We overexpressed Morgana in BT549, as requested by the Reviewer. BT549 overexpressing Morgana invaded more than control cells in an *in vitro* invasion assay and, strikingly, formed lung metastasis in NSG mice. The results are shown in Supplementary Fig. 1d-g (page 6).

There are other comments by figures.

Figure 1: The authors describe MDA-231 metastasis to the liver, heart and kidney, these sites are not well-described sites of metastasis. The authors need to comment since this is a new description of this tumor spreading to these unusual sites.

The formation of metastatic lesions in liver, kidney and heart of NSG mice from MDA-MB-231 primary tumours, albeit rarely reported, have been described previously^{14,15,16}. We added these references at page 5.

Figure 2: It is unclear if Morgana is always internal or does the protein get externalized/secreted? In C, D, include MMP-2 and Morgana for both.

As the Reviewer guessed, Morgana, as other chaperone proteins (Hsp90, Hsp70, etc) can be secreted by cancer cells. We added a Morgana and MMP-2 Western blot analysis in Fig. 2, as requested. We added a comment in the corresponding Figure legend.

Quantify metastatic burden in e.

We quantified the metastatic burden by measuring the area of lung metastasis and calculating the percentage of metastasis area on total lung area in experiments performed on MDA-MB-231, BT549, 4T1 and E0771 (Fig. 1h,m, Supplementary Fig. 1g, and Fig. 8a,b).

In G, h, the authors used TNF- α in the assays, the same should be the same for e and f studies.

For all the NF- κ B target gene expression analysis, we obtained significant results with or without TNF- α treatment. In particular, we showed untreated cells in Fig. 2g,h and we added results of TNF- α treated cells in Fig. 3c,d and Supplementary Fig. 4.

Figure 3: show SP-1 in c, d

We added luciferase analysis for AP1 activity in MCF10A (Supplementary Fig. 3b) and MCF7 cells (Fig. 3b).

Figure 4: repeat western blots for clarity in H and I.

We repeated immunoprecipitation and Western blot analysis in Fig. 4h and we improved Western blot in Fig. 4i.

Figure 5: the survival data needs to be consistent for a and b with the same Y axis blots.

We modified Fig. 5 in accordance with the Reviewer's suggestion (see Fig. 5a,b).

Figure 6: include ShMORG2 for a and for e. Need to be consistent for g and h, need CCL5, IL-6 for both tumor cell types.

We added a second shRNA against Morgana for all the experiments performed on 4T1 cells (Supplementary Fig. 8, Supplementary Fig. 9b and Fig. 6,7,8). We standardized the NF- κ B target genes analyzed by Real-time PCR as suggested by the Reviewer (Supplementary Fig. 8g,h).

Where is TNFa here

We added results for both 4T1 and E0771 treated with TNF- α (Supplementary Fig. 8g,h).

Figure 7: for cell characteristics, the authors should consider B cells, NK cells, dendritic cells, endothelial cells in these studies.

We are grateful to the Reviewer for this suggestion. Very interestingly, the analysis suggests a role for Morgana in inhibiting tumour recognition by NK cells. In fact, in very early phases of tumour growth, when empty and shMorgana tumours had a similar size and neutrophil recruitment was comparable, NK cells present in the tumours were significantly different (Fig.7a-c and Supplementary Fig.10d-f). In particular, primary tumours in which Morgana has been downregulated showed higher levels of infiltrating NK cells. This suggests that Morgana plays a role in escaping NK mediated immunosurveillance. The ability of NK cells to recognize and kill cancer cells depends on the balance between activating or inhibitory signals. MHC class I receptors on cancer cells bind to NK receptors and potentially inhibit their activation¹³. The high recruitment of NK cells in tumours in which Morgana has been silenced may depend on the downregulation of MHC class I gene expression, as highlighted by microarray and/or Real-time PCR analysis on MDA-MB-231, 4T1 and E0771 cells (Fig.7d-f). This issue has been discussed at page 16-17 of the Discussion section.

As requested, we also analyzed B cells, dendritic cells and endothelial cells and no differences were observed in tumours silenced for Morgana compared with controls at all time points (Fig.6b,g, Fig.7b, Fig.8c, e, Supplementary Fig.9d, Supplementary Fig. 10b, e and Supplementary Fig. 11c, d).

The pre-metastatic niche work is very fascinating and should be further explored. How do the neutrophils get recruited to the lungs. There are previous publications on MMP-9 in the pre-metastatic niche, does the authors observe a similar increase in MMP-9? What about IL-6 in the pre-metastatic niche. What happens to ECM changes which can be affected by MMP-9.

We improved our studies on the pre-metastatic niche (PMN), as requested by the Reviewer. Many factors has been described to have an important role in creating PMN, by recruiting immune cells and by rendering PMN hospitable for cancer cells. We analyzed the expression of neutrophil-attracting chemokines (CXCL1, CXCL2, CXCL5, CXCL12), pre-metastatic niche markers (IL-1 β , G-CSF and MMP9), pro-inflammatory mediators (S100A8 and S100A9) and ECM related factors (fibronectin and LOX) by Real-Time PCR. While in lungs of mice injected with control cells the expression of these genes increased consistently, in mice carrying shMorgana tumours they do not differ from tumour-free mice (Fig. 8g). The analysis was performed when shMorgana and control tumours had comparable sizes. These data indicate that the absence of Morgana in breast cancer cells totally impair the establishment of the lung PMN.

References

1. Heiser LM, *et al.* Subtype and pathway specific responses to anticancer compounds in breast cancer. *Proc Natl Acad Sci U S A* **109**, 2724-2729 (2012).
2. Fusella F, *et al.* Morgana acts as a proto-oncogene through inhibition of a ROCK-PTEN pathway. *J Pathol*, (2014).
3. Ferretti R, *et al.* Morgana/chp-1, a ROCK inhibitor involved in centrosome duplication and tumorigenesis. *Dev Cell* **18**, 486-495 (2010).
4. Manuel Iglesias J, *et al.* Mammosphere formation in breast carcinoma cell lines depends upon expression of E-cadherin. *PLoS One* **8**, e77281 (2013).
5. Rinkenbaugh AL, Baldwin AS. The NF-kappaB Pathway and Cancer Stem Cells. *Cells* **5**, (2016).
6. Kendellen MF, Bradford JW, Lawrence CL, Clark KS, Baldwin AS. Canonical and non-canonical NF-kappaB signaling promotes breast cancer tumor-initiating cells. *Oncogene* **33**, 1297-1305 (2014).
7. Shostak K, Chariot A. NF-kappaB, stem cells and breast cancer: the links get stronger. *Breast Cancer Res* **13**, 214 (2011).
8. Liu M, *et al.* The canonical NF-kappaB pathway governs mammary tumorigenesis in transgenic mice and tumor stem cell expansion. *Cancer Res* **70**, 10464-10473 (2010).
9. Jia D, *et al.* beta-Catenin and NF-kappaB co-activation triggered by TLR3 stimulation facilitates stem cell-like phenotypes in breast cancer. *Cell Death Differ* **22**, 298-310 (2015).
10. Duru N, Candas D, Jiang G, Li JJ. Breast cancer adaptive resistance: HER2 and cancer stem cell repopulation in a heterogeneous tumor society. *J Cancer Res Clin Oncol* **140**, 1-14 (2014).
11. Dong H, *et al.* Breast Cancer MDA-MB-231 Cells Use Secreted Heat Shock Protein-90alpha (Hsp90alpha) to Survive a Hostile Hypoxic Environment. *Sci Rep* **6**, 20605 (2016).
12. Xia Y, Shen S, Verma IM. NF-kappaB, an active player in human cancers. *Cancer Immunol Res* **2**, 823-830 (2014).
13. Morvan MG, Lanier LL. NK cells and cancer: you can teach innate cells new tricks. *Nat Rev Cancer* **16**, 7-19 (2016).
14. Iorns E, *et al.* A new mouse model for the study of human breast cancer metastasis. *PLoS One* **7**, e47995 (2012).
15. Milsom CC, Lee CR, Hackl C, Man S, Kerbel RS. Differential post-surgical metastasis and survival in SCID, NOD-SCID and NOD-SCID-IL-2Rgamma(null) mice with parental and subline variants of human breast cancer: implications for host defense mechanisms regulating metastasis. *PLoS One* **8**, e71270 (2013).
16. Puchalapalli M, *et al.* NSG Mice Provide a Better Spontaneous Model of Breast Cancer Metastasis than Athymic (Nude) Mice. *PLoS One* **11**, e0163521 (2016).

REVIEWERS' COMMENTS:

Reviewer #1 (Remarks to the Author):

The authors have addressed my concerns successfully. The work is interesting and novel (regarding a regulatory component in the NF- κ B signaling pathway). The link with cancer is also important.

Reviewer #2 (Remarks to the Author):

I'm very pleased with the revised version of this manuscript. The authors have addressed all my concerns. I'm looking forward to see this solid work describing the novel role of Morgana in breast cancer growth and progression, published in Nature Communications.

Reviewer #3 (Remarks to the Author):

In the revised version of the manuscript by F. Fusella et al, the authors have answered all the concerns of this reviewer. The work has greatly improved which has strengthened the novelty of their findings. There are no additional concerns.